# CSER: Communication-efficient SGD with Error Reset

**Cong Xie** [* 1,2]
cx2@illinois.edu

**Shuai Zheng** [2]
shzheng@amazon.com

**Oluwasanmi Koyejo** [1]
sanmi@illinois.edu

**Indranil Gupta** [1]
indy@illinois.edu

**Mu Li** [2]
mli@amazon.com

**Haibin Lin** [2]
haibin.lin.aws@gmail.com

[1] Department of Computer Science
University of Illinois Urbana-Champaign

[2] Amazon Web Services

## Abstract

The scalability of Distributed Stochastic Gradient Descent (SGD) is today limited by communication bottlenecks. We propose a novel SGD variant: Communication-efficient SGD with Error Reset, or CSER. The key idea in CSER is first a new technique called "error reset" that adapts arbitrary compressors for SGD, producing bifurcated local models with periodic reset of resulting local residual errors. Second we introduce partial synchronization for both the gradients and the models, leveraging advantages from them. We prove the convergence of CSER for smooth non-convex problems. Empirical results show that when combined with highly aggressive compressors, the CSER algorithms accelerate the distributed training by nearly $10\times$ for CIFAR-100, and by $4.5\times$ for ImageNet.

## 1 Introduction

In recent years, the sizes of both machine-learning models and datasets have been increasing rapidly. To accelerate the training, it is common to distribute the computation on multiple machines. We focus on Stochastic Gradient Descent (SGD). SGD and its variants are commonly used for training large-scale deep neural networks. A common way to distribute SGD is to synchronously compute the gradients at multiple worker nodes, and then aggregate the global average. This is akin to single-threaded SGD with large mini-batch sizes [5, 27–29]. Increasing the number of workers is attractive because it holds the potential to reduce training time. However, more workers also means more communication, and overwhelmed communication links hurt scalability.

The state-of-the-art work in communication-efficient SGD is called QSparse-local-SGD [3], which combines two prevailing techniques: message compression and infrequent synchronization. Message compression methods use compressors such as quantization [2, 4, 9, 17, 21, 26, 33] and sparsification [1, 8, 20] to reduce the number of bits in each synchronization round. This necessitates error feedback (EF-SGD) [9, 33] to correct for the residual errors incurred by the compressors, and to guarantee theoretical convergence. On the other hand, infrequent synchronization methods such as local SGD [13, 19, 25, 30, 31] would decrease the overall number of synchronization rounds. The former, QSparse-local-SGD, periodically synchronizes the model parameters like local SGD, and compresses the synchronization messages to further reduce the communication overhead. Similar to EF-SGD, it also uses error feedback to correct for the residual errors of compression.

---

[*]The work was done when Cong Xie was a (part-time) intern in Amazon Web Services.

QSparse-local-SGD reduces more bidirectional communication overhead (in both aggregation and broadcasting) than its ancestors EF-SGD and local SGD. However, it also inherits weaknesses from both ancestor algorithms, especially when compression ratios are increased. For instance, our experiments reveal that QSparse-local-SGD fails to converge at a compression ratio of $256\times$.

In this paper, we introduce a new algorithm called Communication-efficient SGD with Error Reset, or CSER. The key idea in CSER is a new technique called *error reset* that corrects for the local model using the compression errors, and we show this converges better than the error feedback technique used in QSparse-local-SGD. On top of the error reset, we also introduce partial synchronization, leveraging advantages from both gradient and model synchronizations. These two techniques together allow the proposed method to scale up the compression ratio to as high as $1024\times$ and significantly outperform the existing approaches.

The main contributions of our paper are as follows:

- We propose a novel communication-efficient SGD algorithm, called Communication-efficient SGD with Error Reset (CSER) as well as its variant with Nesterov's momentum [15]. CSER includes a new technique that adapts arbitrary compressors for SGD, and achieves better convergence than the baselines when aggressive compressors are used.

- We add a second compressor to partially synchronize the gradients between the resets of errors on local models. We show that tuning the compression ratios between the gradient synchronization and model synchronization improves the convergence.

- We show empirically that with appropriate compression ratios, CSER accelerates distributed training by nearly $10\times$ for CIFAR-100, and by $4.5\times$ for ImageNet.

## 2 Related work

Recently, Basu et al. (2019) proposed QSparse-local-SGD, which combines local SGD [13, 19, 25, 30, 31] and EF-SGD [9, 33], and reduces more communication overhead than any single one of them. The detailed algorithm of QSparse-local-SGD is shown in Algorithm 1. In the algorithm, $x_{i,t}$ is the local model on the $i$th worker in the $t^{\text{th}}$ iteration, and $\hat{x}_t$ is the globally synchronized model in the $t^{\text{th}}$ iteration. Note that $\hat{x}_t$ is always the same across different workers, which is used to track the synchronized part of the local model $x_{i,t}$. In Line 9, the local residual error $e_{i,t-1}$ from the previous synchronization round is added to the accumulated local update $x_{i,t-\frac{1}{2}} - \hat{x}_{t-1}$. In Line 10, the message $p_{i,t}$ is compressed into $p'_{i,t}$. Line 11 produces the residual error $e_{i,t}$ of the compression, and synchronizes the compressed messages. Finally, in Line 12, the synchronized update is accumulated to the local models. When $H = 1$, the algorithm is reduced to EF-SGD. If the message compression in Line 10 is an identity mapping (i.e., $\mathcal{C}_1(p_{i,t}) = p_{i,t}$), then the algorithm is reduced to local SGD.

---

**Algorithm 1** Qsparse-local-SGD

1: **Input**: $\mathcal{C}_1$ - compressor, $H > 0$ - synchronization interval
2: Initialize $x_{i,0} = \hat{x}_0 \in \mathbb{R}^d, e_{i,t} = \mathbf{0}, \forall i \in [n]$
3: **for all** iteration $t \in [T]$ **do**
4:     **for all** Workers $i \in [n]$ in parallel **do**
5:         $x_{i,t-\frac{1}{2}} \leftarrow x_{i,t-1} - \eta \nabla f(x_{i,t-1}; z_{i,t})$
6:         **if** $\mod (t, H) \neq 0$ **then**
7:             $x_{i,t} \leftarrow x_{i,t-\frac{1}{2}}, \quad \hat{x}_t \leftarrow \hat{x}_{t-1}, \quad e_{i,t} \leftarrow e_{i,t-1}$
8:         **else**
9:             $p_{i,t} \leftarrow e_{i,t-1} + x_{i,t-\frac{1}{2}} - \hat{x}_{t-1}$
10:           $p'_{i,t} \leftarrow \mathcal{C}_1(p_{i,t})$
11:           $e_{i,t} \leftarrow p_{i,t} - p'_{i,t}, \quad \bar{p}'_t \leftarrow \frac{1}{n} \sum_{i \in [n]} p'_{i,t}$           ▷ Synchronization
12:           $x_{i,t} \leftarrow \hat{x}_{t-1} + \bar{p}'_t, \quad \hat{x}_t \leftarrow \hat{x}_{t-1} + \bar{p}'_t$
13:         **end if**
14:     **end for**
15: **end for**

---

In QSparse-local-SGD, the residual error $e_{i,t}$ are left aside from gradient computation in the $H$ local steps. It is not applied to the local models until synchronization. Thus, the staleness of the residual error is at least $H$ iterations. Such staleness in the error feedback grows with the compression ratio $R_{\mathcal{C}_1}$ and synchronization interval $H$, which causes potential convergence issues when the overall compression ratio $R_{\mathcal{C}} = R_{\mathcal{C}_1} \times H$ is large. As a result, we observe bad convergence of QSparse-local-SGD in our experiments when $R_{\mathcal{C}} \geq 256$. On the other hand, EF-SGD uses $H = 1$ but also shows relatively bad performance using random sparsifiers in both previous work [20] and our experiments. When $\mathcal{C}_1$ is an identity mapping, QSparse-local-SGD reduces to local SGD, in which the differences between the local models still grow with the synchronization interval $H$, resulting in slow convergence when $H$ is large.

## 3 Methodology

We consider the following optimization problem with $n$ workers: $\min_{x \in \mathbb{R}^d} F(x)$, where $F(x) = \frac{1}{n} \sum_{i \in [n]} F_i(x) = \frac{1}{n} \sum_{i \in [n]} \mathbb{E}_{z_i \sim \mathcal{D}_i} f(x; z_i), \forall i \in [n]$, and $z_i$ is sampled from the local data $\mathcal{D}_i$ on the $i$th device. Furthermore, we assume $\mathcal{D}_i \neq \mathcal{D}_j, \forall i \neq j$.

We solve this optimization problem using distributed SGD and its variants. To reduce the communication overhead, we compress the messages via $\delta$-approximate compressors.

**Definition 1** (Karimireddy et al. [9]). *An operator $\mathcal{C} : \mathbb{R}^d \to \mathbb{R}^d$ is a $\delta$-approximate compressor for $\delta \in [0, 1]$ if $\|\mathcal{C}(v) - v\|^2 \leq (1 - \delta)\|v\|^2, \forall v \in \mathbb{R}^d$.*

Note that in the original definition of the compressor, it is required that $\delta \in (0, 1]$. In this paper, we extend this assumption by allowing $\delta = 0$, where $\mathcal{C}(v) = 0$ in some cases.

### 3.1 Communication-efficient SGD with error reset

We propose a new procedure to apply arbitrary $\delta$-approximate compressors to distributed SGD, which achieves good accuracy when using aggressive compressors and fixes the potential convergence issues of QSparse-local-SGD. Our procedure directly applies the residuals to the local models, then uses the local models to compute the gradients in the next iteration, which results in bifurcated local models similar to local SGD. Observe that in contrast, QSparse-local-SGD has the local models fully synchronized across the workers after each synchronization round, and puts the residuals aside from the gradient computation during the local updates.

The proposed algorithm periodically resets the errors that are locally accumulated on the models on workers. Thus, we denote this algorithm as communication-efficient SGD with error reset (CSER). In Table 1, we summarize the techniques used in CSER, and how this differs from existing work.

Table 1: Our approach (CSER) vs. Existing Techniques (EF-SGD, QSparse-local-SGD)

| | Message compression | Infrequent synchronization | Momentum with provable convergence | Aggressive compressor | Error reset |
|---|---|---|---|---|---|
| EF-SGD | ✓ | | ✓ | | |
| QSparse-local-SGD | ✓ | ✓ | | | |
| **CSER (this paper)** | ✓ | ✓ | ✓ | ✓ | ✓ |

In Algorithm 3, we define a sub-routine which partially synchronizes the tensors. Given any compressor $\mathcal{C}$, on any worker $i$, the sub-routine takes the average only over the compressed part of the messages, and locally combines the residual with the averaged value.

Applying the sub-routine (Algorithm 3) to distributed SGD, we propose a new algorithm with two arbitrary compressors: $\mathcal{C}_1$ and $\mathcal{C}_2$, with approximation factor $\delta_1 \in (0, 1]$ and $\delta_2 \in [0, 1]$, respectively. The detailed algorithm is shown in Algorithm 2. In the algorithm, $x_{i,t}$ is the local model on the $i$th worker in the iteration $t$. In Line 11 and 12, the first compressor $\mathcal{C}_1$ flushes the local error $e_{i,t}$ by partial synchronization, i.e., the local errors are (partially) reset for every $H$ iterations, which is similar to QSparse-local-SGD. Between the error-reset rounds, we add a second compressor $\mathcal{C}_2$ to partially synchronize the gradients (Line 6), and accumulate both the synchronized values and the residuals to the local models (Line 7).

The locally accumulated residual error $e_{i,t}$ maintains the differences between the local models, which causes additional noise to the convergence. Formally, we have

**Lemma 1.** *$x_{i,t} - e_{i,t}$ is the same across different workers: $x_{i,t} - e_{i,t} = x_{j,t} - e_{j,t}, \forall i,j \in [n], t$.*

Different from the error feedback of QSparse-local-SGD, the error reset of CSER applies the residual errors immediately to the local models without delay, and thus avoids the issue of staleness and improves the convergence. Additionally, by utilizing both gradient and model synchronization, and balancing the communication budget between them, CSER achieves a better trade-off between the accuracy and the reduction of bidirectional communication. When all the budget is on $\mathcal{C}_1$, the local models bifurcate too much, which leads to bad accuracy as local SGD. Instead, we trade off some budget of $\mathcal{C}_1$ for the partial synchronization of gradients with $\mathcal{C}_2$, thus mitigate the weaknesses. Furthermore, with specially designed sparsifiers, the proposed algorithms no longer need to maintain the variables $e_{i,t}$. The resultant implementation reduces the memory footprint and the corresponding overhead of memory copy. Details are introduced in Section 3.3 and Appendix A.4.

Algorithm 2 allows great freedom in tuning the two different compressors $\mathcal{C}_1$ and $\mathcal{C}_2$, as well as the error-reset interval $H$. By specifying the hyperparameters, we recover some important special cases of CSER. Some existing approaches are similar to these special cases, though the differences often turn out to be important. The details can be found in Appendix A.

---

**Algorithm 2** CSER

1: **Input**: $\mathcal{C}_1, \mathcal{C}_2$ - compressors, $H > 0$ - error-reset interval
2: Initialize $x_{i,0} = \hat{x}_0 \in \mathbb{R}^d, e_{i,0} = \mathbf{0}, \forall i \in [n]$
3: **for all** iteration $t \in [T]$ **do**
4:     **for all** Workers $i \in [n]$ in parallel **do**
5:         $g_{i,t} \leftarrow \nabla f(x_{i,t-1}; z_{i,t}), z_{i,t} \sim \mathcal{D}_i$
6:         $g'_{i,t}, r_{i,t} \leftarrow PSync(g_{i,t}, \mathcal{C}_2)$
7:         $x_{i,t-\frac{1}{2}} \leftarrow x_{i,t-1} - \eta g'_{i,t}, \quad e_{i,t-\frac{1}{2}} \leftarrow e_{i,t-1} - \eta r_{i,t}$
8:         **if** $\mod (t, H) \neq 0$ **then**
9:             $x_{i,t} \leftarrow x_{i,t-\frac{1}{2}}, \quad e_{i,t} \leftarrow e_{i,t-\frac{1}{2}}$
10:        **else**
11:            $e'_{i,t-\frac{1}{2}}, e_{i,t} \leftarrow PSync(e_{i,t-\frac{1}{2}}, \mathcal{C}_1)$
12:            $x_{i,t} \leftarrow x_{i,t-\frac{1}{2}} - e_{i,t-\frac{1}{2}} + e'_{i,t-\frac{1}{2}}$
13:        **end if**
14:    **end for**
15: **end for**

---

**Algorithm 3** Partial Synchronization (PSync)

1: **Input**: $v_i \in \mathbb{R}^d$, $\mathcal{C}$ - compressor
2: **function** PSYNC$(v_i, \mathcal{C})$
3:     On worker $i$:
4:         $v'_i = \mathcal{C}(v_i)$
5:         $r_i = v_i - v'_i$
6:         Partial synchronization:
7:             $\bar{v}' = \frac{1}{n} \sum_{i \in [n]} v'_i$
8:             $v'_i = \bar{v}' + r_i$
9:         **return** $v'_i, r_i$
10: **end function**

---

## 3.2 Momentum variant

Nesterov's momentum [15] is a variant of SGD that has been widely used to accelerate the convergence. Sutskever et al. (2013) show that Nesterov's momentum can be expressed in terms of a classic momentum update as:

$$
\begin{aligned}
m_t &= \beta m_{t-1} + g_t, \\
x_t &= x_{t-1} - \eta(\beta m_t + g_t),
\end{aligned}
$$

where $\beta$ is the momentum parameter, $g_t$ is the gradient. Nesterov's momentum moves the model parameters in the direction of the accumulated gradient. Very recently, Zheng et al. (2019) incorporate Nesterov's momentum into EF-SGD with bidirectional communication and obtains faster convergence. In this section, we introduce M-CSER that adopts Nesterov's momentum in CSER. Compared to Algorithm 2, the momentum variant simply adds momentum to the gradients before applying the second compressor $\mathcal{C}_2$, as shown in Algorithm 4.

## 3.3 Globally-randomized blockwise sparsifier (GRBS)

There are two sparsifiers widely used with SGD: random-$k$ and top-$k$ sparsifiers. Random-$k$ sparsifiers select random elements for synchronization, while top-$k$ sparsifiers select the most significant elements. Top-$k$ sparsifiers typically achieve better convergence [20], but also incur heavier overhead.

---

**Algorithm 4** Distributed Momentum SGD with Error-Reset (M-CSER, implementation I)

---
1: **Input**: $\mathcal{C}_1, \mathcal{C}_2$ - compressors, $H > 0$ - synchronization interval
2: Initialize $x_{i,0} = \hat{x}_0 \in \mathbb{R}^d, e_{i,0} = \mathbf{0}, m_{i,0} = \mathbf{0}, \forall i \in [n]$
3: **for all** iteration $t \in [T]$ **do**
4:      **for all** Workers $i \in [n]$ in parallel **do**
5:          $g_{i,t} \leftarrow \nabla f(x_{i,t-1}; z_{i,t}), z_{i,t} \sim \mathcal{D}_i$
6:          $m_{i,t} \leftarrow \beta m_{i,t-1} + g_{i,t}$
7:          $p_{i,t} \leftarrow \eta(\beta m_{i,t} + g_{i,t})$
8:          $p'_{i,t}, r_{i,t} \leftarrow PSync(p_{i,t}, \mathcal{C}_2)$
9:          $x_{i,t-\frac{1}{2}} \leftarrow x_{i,t-1} - p'_{i,t}, \quad e_{i,t-\frac{1}{2}} \leftarrow e_{i,t-1} - r_{i,t}$
10:          **if** $\mod(t, H) \neq 0$ **then**
11:              $x_{i,t} \leftarrow x_{i,t-\frac{1}{2}}, \quad e_{i,t} \leftarrow e_{i,t-\frac{1}{2}}$
12:          **else**
13:              $e'_{i,t-\frac{1}{2}}, e_{i,t} \leftarrow PSync(e_{i,t-\frac{1}{2}}, \mathcal{C}_1)$                             ▷ error reset
14:              $x_{i,t} \leftarrow x_{i,t-\frac{1}{2}} - e_{i,t-\frac{1}{2}} + e'_{i,t-\frac{1}{2}}$
15:          **end if**
16:      **end for**
17: **end for**

---

In this paper, we use a blockwise random sparsifier with synchronized random seed, which is also mentioned in [23].

**Definition 2.** *(Globally-Randomized Blockwise Sparsifier, GRBS) Given any tensors $v_i \in \mathbb{R}^d, i \in [n]$ distributed on the $n$ workers, the compression ratio $R_{\mathcal{C}}$, and the number of blocks $B$, GRBS partitions each $v_i$ into $B$ blocks. In each iteration, GRBS globally picks $\frac{B}{R_{\mathcal{C}}}$ random blocks for synchronization, and GRBS is a $1/R_{\mathcal{C}}$-approximate compressor in expectation.*

Compared to the other compressors, GRBS has the following advantages:

- **Adaptivity to AllReduce and parameter server:** Due to the synchronized random seed, different workers always choose the same blocks for synchronization. Thus, GRBS is compatible with AllReduce [18, 24] and parameter server [7, 11, 12]. Other compressors such as random sparsifier and quantization cannot be directly employed with Allreduce or parameter server since their compressed gradients cannot be directly summed without first be decompressed.

- **Less memory footprint:** With GRBS, CSER can further reduce the memory footprint and the corresponding overhead of memory copy. Implementation details are shown in Appendix A.4.

Although GRBS has less communication and computation overhead, it is too aggressive for the existing algorithms such as QSparse-local-SGD when we consider a large $R_{\mathcal{C}}$. In Section 5.2, we show that CSER improves the convergence when the overall compression ratio is as large as $1024\times$.

## 4 Convergence analysis

In this section, we present the convergence guarantees of CSER.

### 4.1 Assumptions

First, we introduce some assumptions for our convergence analysis.

**Assumption 1.** *$F_i(x), \forall i \in [n]$ are $L$-smooth: $F_i(y) - F_i(x) \leq \langle \nabla F_i(x), y - x \rangle + \frac{L}{2} \|y - x\|^2, \forall x, y$.*

**Assumption 2.** *For any stochastic gradient $g_{i,t} = \nabla f(x_{i,t-1}; z_{i,t}), z_{i,t} \sim \mathcal{D}_i$, we assume bounded variance and expectation: $\mathbb{E}[\|g_{i,t} - \nabla F_i(x_{i,t-1})\|^2] \leq V_1, \|\mathbb{E}[g_{i,t}]\|^2 \leq V'_1, \forall i \in [n], t \in [T]$. Furthermore, gradients from different workers are independent from each other.*

Note that this implies the bounded second moment: $\mathbb{E}[\|g_{i,t}\|^2] \leq V_2 \equiv V_1 + V'_1, \forall i \in [n], t \in [T]$.

**Assumption 3.** *There exists at least one global minimum $x_*$, where $F(x_*) \leq F(x), \forall x$.*

## 4.2 Main results

Based on the assumptions above, we have the following convergence guarantees. The detailed proof can be found in Appendix B. To analyze the proposed algorithms, we introduce auxiliary variables:

$$\bar{x}_t = \frac{1}{n} \sum_{i \in [n]} x_{i,t}.$$

We show that the sequence $\{\bar{x}_{t-1} : t \in [T]\}$ converge to a critical point.

**Theorem 1.** *Taking $\eta \leq \frac{1}{L}$, after $T$ iterations, Algorithm 2 (CSER) has the following error bound:*

$$\frac{1}{T} \sum_{t=1}^{T} \mathbb{E}\left[\|\nabla F(\bar{x}_{t-1})\|^2\right] \leq \mathcal{O}\left(\frac{1}{\eta T}\right) + \mathcal{O}\left(\frac{\eta L V_1}{n}\right) + 2\left[\frac{4(1-\delta_1)}{\delta_1^2} + 1\right](1-\delta_2)\eta^2 H^2 L^2 V_2.$$

The following corollary shows that CSER has a convergence rate of $\mathcal{O}\left(\frac{1}{\sqrt{nT}}\right)$, leading a linear speedup using more workers.

**Corollary 1.** *Taking $\eta = \min\left\{\frac{\gamma}{\sqrt{T/n} + \left[4(1-\delta_1)/\delta_1^2 + 1\right]^{1/3} 2^{1/3}(1-\delta_2)^{1/3} H^{2/3} T^{1/3}}, \frac{1}{L}\right\}$ for some $\gamma > 0$, after $T \gg n$ iterations, Algorithm 2 (CSER) converges to a critical point:*

$$\frac{1}{T} \sum_{t=1}^{T} \mathbb{E}\left[\|\nabla F(\bar{x}_{t-1})\|^2\right] \leq \mathcal{O}\left(\frac{1}{\sqrt{nT}}\right).$$

To compare the error bounds between CSER and QSparse-local-SGD, we quote the following results (reformatted to match the notations in this paper) from Theorem 1 of [3] without proof.

**Lemma 2.** *[3] Taking $\eta \leq \frac{1}{2L}$, QSparse-local-SGD has the error bound:*

$$\frac{1}{T} \sum_{t=1}^{T} \mathbb{E}\left[\|\nabla F(\bar{x}_{t-1})\|^2\right] \leq \mathcal{O}\left(\frac{1}{\eta T}\right) + \mathcal{O}\left(\frac{\eta L V_1}{n}\right) + 8\left[\frac{4(1-\delta_1^2)}{\delta_1^2} + 1\right]\eta^2 H^2 L^2 V_2.$$

Comparing Lemma 2 with Theorem 1, CSER shows a better error bound than QSparse-local-SGD.

**Remark 1.** *Taking $\delta_2 = 0$, and the same $\delta_1$ as QSparse-local-SGD, CSER reduces the compression error to $2\left[\frac{4(1-\delta_1)}{\delta_1^2} + 1\right]\eta^2 H^2 L^2 V_2$, compared to $8\left[\frac{4(1-\delta_1^2)}{\delta_2^2} + 1\right]\eta^2 H^2 L^2 V_2$ of QSparse-local-SGD. Ignoring the constant factors, the error caused by $\mathcal{C}_1$ is reduced from $\frac{4(1-\delta_1^2)}{\delta_1^2}$ to $\frac{4(1-\delta_1)}{\delta_1^2}$.*

Though the eliminated factor $(1 + \delta_1)$ seems small, it could lead to significant gaps in the convergence. For example, taking $H = 8$ and $\delta_1 = 1/2$, CSER reduces the compression error from 832 to 576.

Furthermore, note that error reset utilizes the local residuals $e_{i,t}$ in a way different from error feedback. Diving deep into the proofs, we find that their compression errors have different sources.

**Remark 2.** *The compression error term of the error reset comes from the variance of the local models: $\frac{1}{n} \sum_{i \in [n]} \left\| \frac{1}{n} \sum_{j \in [n]} x_{j,t} - x_{i,t} \right\|^2$, which equals to $\frac{1}{n} \sum_{i \in [n]} \left\| \frac{1}{n} \sum_{j \in [n]} e_{j,t} - e_{i,t} \right\|^2 \leq \frac{1}{n} \sum_{i \in [n]} \|e_{i,t}\|^2$ using Lemma 1. This variance vanishes when $n = 1$. However, for error feedback, the compression error is bounded by $\frac{1}{n} \sum_{i \in [n]} \|e_{i,t}\|^2$, which does not vanish when $n = 1$.*

The remark above shows that error reset always has a smaller error bound compared to error feedback. Especially, when using a single worker, CSER is equivalent to SGD with no compression error, while QSparse-local-SGD has the compressor error even using a single worker with $H = 1$.

Besides error reset, CSER introduces partial synchronization for both the gradients and the models. By carefully tuning the communication budget between them, the convergence can be improved.

For example, assume that we use $\mathcal{C}_{GRBS}$ introduced in Definition 2 that has a compression ratio $R_{\mathcal{C}}$ and satisfies $\mathbb{E}[\|\mathcal{C}_{GRBS}(v) - v\|_2^2] \leq (1 - \frac{1}{R_{\mathcal{C}}})\|v\|_2^2$. If we put all the budget to

model synchronization, and take $H = 4$, $\delta_1 = 1/3$, $\delta_2 = 0$, the compression error is $\left[\frac{4(1-\delta_1)}{\delta_1^2} + 1\right]\eta^2 H^2 L^2 V_2 = 400\eta^2 L^2 V_2$. However, if we move some budget to gradient synchronization and take $H = 12, \delta_1 = 7/8, \delta_2 = 1/96$, the overall compression budget remains the same, but the error term is reduced to less than $236\eta^2 L^2 V_2$.

We also establish the convergence analysis for CSER with Nesterov's momentum.

**Theorem 2.** *Taking $\eta \leq \min\{\frac{1}{2}, \frac{1-\beta}{2L}\}$, after $T$ iterations, Algorithm 4 (M-CSER) has the following error bound:*

$$\frac{1}{T}\sum_{t=1}^{T}\mathbb{E}\left[\|\nabla F(\bar{x}_{t-1})\|^2\right] \leq \frac{2(1-\beta)\left[F(\bar{x}_0) - F(x_*)\right]}{\eta T}$$

$$+ \frac{\eta^2\beta^4 L^2 V_2}{(1-\beta)^4} + \frac{\eta L V_1}{n(1-\beta)} + \left(\frac{4(1-\delta_1)}{\delta_1^2} + 1\right)\frac{2(1-\delta_2)\eta^2 H^2 L^2 V_2}{(1-\beta)^2}.$$

Note that larger $\beta$ leads to faster escape from the initial point, but worse asymptotic performance.

**Corollary 2.** *Taking $\eta = \min\left\{\frac{\gamma}{\sqrt{T/n}+\left[2\left(4(1-\delta_1)/\delta_1^2+1\right)(1-\delta_2)H^2+1\right]^{1/3}T^{1/3}}, \frac{1}{2}\right\}$ for some $\gamma > 0$, after $T \geq \frac{4\gamma^2 L^2 n}{(1-\beta)^2}$ iterations, Algorithm 4 (M-CSER) converges to a critical point:*

$$\frac{1}{T}\sum_{t=1}^{T}\mathbb{E}\left[\|\nabla F(\bar{x}_{t-1})\|^2\right] \leq \mathcal{O}\left(\frac{1}{\sqrt{nT}}\right).$$

Similar to CSER, Corollary 2 shows that Algorithm 4 (M-CSER) converges to a critical point at the rate $\mathcal{O}\left(\frac{1}{\sqrt{nT}}\right)$. Increasing the number of workers $n$ accelerates the convergence.

## 5 Experiments

In this section, we report the empirical results in a distributed environment.

### 5.1 Evaluation setup

We compare our algorithms with 3 baselines: SGD with full precision (SGD in brief), EF-SGD, and QSparse-local-SGD. We use momentum to accelerate the training in all the experiments, though QSparse-local-SGD with momentum does not have convergence guarantees in its original paper [3].

We conduct experiments on two image classification benchmarks: CIFAR-100 [10], and ImageNet dataset [16], in a cluster of 8 machines where each machine has 1 NVIDIA V100 GPU and up to 10 Gb/s networking bandwidth. Each experiment is repeated 5 times.

For CIFAR-100, we use the wide residual network (Wide-ResNet-40-8, [32]). We set weight decay to 0.0005, momentum to 0.9, and minibatch size to 16 per worker. We decay the learning rates by 0.2 at 60, 120 and 160 epochs, and train for 200 epochs. The initial learning rate is varied in $\{0.05, 0.1, 0.5, 1.0\}$.

For ImageNet, we use a 50-layer ResNet [6]. We set weight decay to 0.0001, momentum to 0.9, and minibatch size to 32 per worker. We use a learning rate schedule consisting of 5 epochs of linear warmup, followed by a cosine-annealing learning-rate decay [14], and train for total 120 epochs. We enumerate the initial learning rates in $\{0.025, 0.05, 0.1, 0.5\}$.

For all the algorithms, we test the performance with different overall compression ratios ($R_{\mathcal{C}}$). We use the globally-randomized blockwise sparsifier (GRBS) as the compressor, as proposed in Section 3.3. Note that CSER has not only two different compressors with compression ratios $R_{\mathcal{C}_1}$ and $R_{\mathcal{C}_2}$ respectively, but also the synchronization interval $H$. The overall compression ratio $R_{\mathcal{C}}$ of CSER is $\frac{1}{1/R_{\mathcal{C}_2}+1/(R_{\mathcal{C}_1}\times H)}$. For QSparse-local-SGD, its overall $R_{\mathcal{C}}$ is $R_{\mathcal{C}_1} \times H$. Note that QSparse-local-SGD is reduced to local SGD when taking $R_{\mathcal{C}_1} = 1$, which is also tested in our experiments. The detailed configurations of of $H$, $R_{\mathcal{C}_1}$, and $R_{\mathcal{C}_2}$ can be found in Appendix C.

Due to brevity we show only high compression ratio results. Appendix D shows further results.

## 5.2 Empirical results

Table 2 presents the test accuracy on CIFAR-100 with various compression ratios. We evaluate not only CSER, but also the other two special cases: CSEA and CSER-PL. The details of the special cases could be found in Appendix A. Note that for CSER, CSER-PL, and QSparse-local-SGD with the same overall $R_{\mathcal{C}}$, the configurations of $H$, $R_{\mathcal{C}_1}$, and $R_{\mathcal{C}_2}$ are not unique. We try multiple configurations and report the ones perform best on the training loss.

Table 2: Testing accuracy (%) on CIFAR-100 with different overall compression ratios ($R_{\mathcal{C}}$). Note that fully synchronous SGD does not have compression, thus $R_{\mathcal{C}} = 1$, and all the other algorithms do not have the fully synchronous cases, thus $R_{\mathcal{C}} \geq 2$.

| Optimizer/ | Baseline | | | Proposed algorithm | | |
| $R_{\mathcal{C}}$ | SGD | EF-SGD | QSparse-local -SGD | CSEA | CSER | CSER-PL |
|---|---|---|---|---|---|---|
| 1 | 87.01±0.11 | | | | | |
| 2 | | 87.20±0.10 | 87.16±0.03 | 87.17±0.21 | **87.47±0.03** | |
| 4 | | 86.97±0.08 | 87.08±0.22 | 87.25±0.23 | 87.22±0.03 | **87.33±0.05** |
| 8 | | 86.61±0.23 | 87.15±0.10 | 87.14±0.05 | 87.09±0.05 | **87.27±0.04** |
| 16 | | 85.69±0.31 | 87.02±0.13 | 87.15±0.09 | **87.28±0.04** | 86.72±0.05 |
| 32 | | 85.17±0.12 | 86.70±0.04 | 86.83±0.20 | 86.90±0.15 | **86.92±0.26** |
| 64 | | 84.65±0.07 | 80.64±0.47 | 86.63±0.16 | 86.78±0.11 | **86.91±0.15** |
| 128 | | 83.50±0.87 | 70.27±2.37 | 86.30±0.15 | **86.81±0.17** | 86.36±0.21 |
| 256 | | 83.92±0.55 | diverge | 86.34±0.20 | **86.68±0.07** | 86.27±0.02 |
| 512 | | 76.05±0.56 | diverge | 85.75±0.34 | **86.20±0.09** | 85.68±0.12 |
| 1024 | | diverge | diverge | 85.13±0.13 | **85.66±0.07** | 84.94±0.37 |

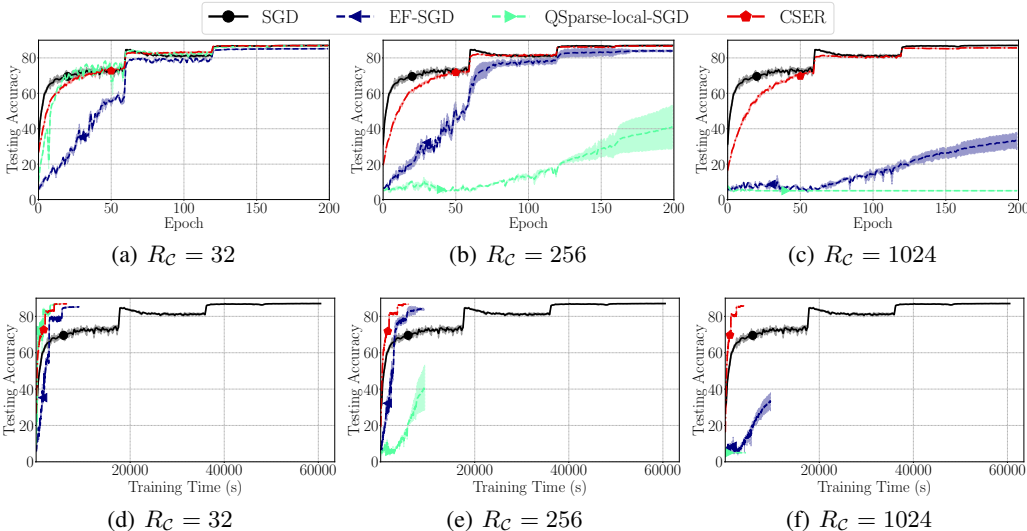

Figure 1: Testing accuracy with different algorithms, for WideResNet-40-8 on CIFAR-100.

In Figure 1 and Figure 2, we show the test accuracy on CIFAR-100 and ImageNet respectively, with the overall compression ratios in $\{32, 256, 1024\}$. Since the experiments on ImageNet are expensive, we do not tune different configurations of compressors ($H, R_{\mathcal{C}_1}, R_{\mathcal{C}_2}$) for each overall $R_{\mathcal{C}}$ on ImageNet, but directly use the best configurations tuned on CIFAR-100.

## 5.3 Discussion

We can see that in all the experiments, with the same compression ratio, CSER shows better performance than the baselines. When the compression ratio is small enough ($\leq 16$), the test accuracy is even better than fully synchronous SGD on CIFAR-100. When $R_{\mathcal{C}} \leq 32$, for CIFAR-100, QSparse-local-SGD has comparable performance to CSER or its special cases. Even with very large

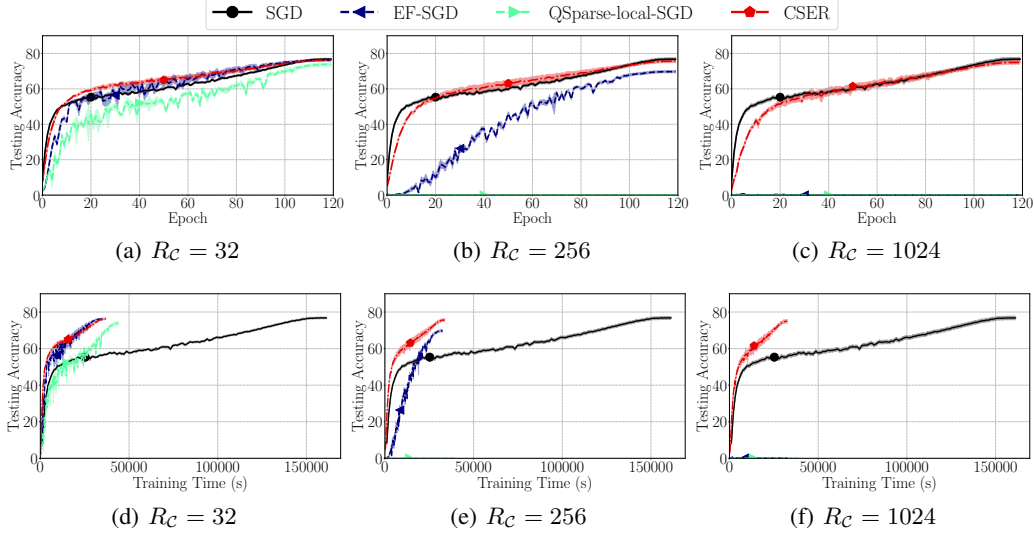

Figure 2: Testing accuracy with different algorithms, for ResNet-50 on ImageNet.

compression ratio ($\approx 256$), the proposed algorithm can achieve comparable accuracy to SGD with full precision. CSER accelerates training by $10\times$ for CIFAR-100, and $4.5\times$ for ImageNet.

Compared to EF-SGD, CSER shows much better performance when the compression ratio is large ($R_\mathcal{C} \geq 64$). We can see that CSER fixes the convergence issue in EF-SGD and QSparse-local-SGD when aggressive compressors are used, as discussed in Section 3.1. For ImageNet with $R_C = 1024$, even if we decrease the learning rates to $0.025$, EF-SGD and QSparse-local-SGD still diverge, while CSER still converges well with even larger learning rates.

Note that in most cases, CSER performs better than CSEA and CSER-PL. The reason is that CSER uses both gradient partial synchronization and model partial synchronization. With finely tuned compression ratios, the local models will not be too far away from each other between the model synchronization rounds, which results in better convergence. Note that although CSEA has slightly worse performance compared to CSER and CSER-PL, it has less hyperparameters to be tuned for the compressors, which is more user-friendly in practice.

# 6 Conclusion

We proposed a novel communication-efficient SGD algorithm called CSER. We introduce error reset and partial synchronization that enable an aggressive compression rate as high as $1024\times$. Theoretically, we show that the proposed algorithm enjoys a linear speedup using more workers. Our empirical results show that the proposed algorithm accelerates the training of deep neural networks. In future work, we will apply our algorithms to other datasets and applications.

## Broader Impact

As this work is mainly algorithmic, the impact is mainly in scientific aspects rather than ethical and societal aspects. Hopefully, our work would enable faster training of machine learning models without regression in accuracy. It would save not only the time but also the expense cost by training large and complex models. On the other hand, there are some related aspects that we have not studied in this work. For example, we do not know how our approaches impact fairness and privacy of the model training, which will be our future work.

## Acknowledgments and Disclosure of Funding

This work was funded in part by the following grants: NSF IIS 1909577, NSF CNS 1908888, NSF CCF 1934986 and a JP Morgan Chase Fellowship, along with computational resources donated by Intel, AWS, and Microsoft Azure.

## Competing Interests

Part-time employment with Google Research Accra (OK).

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
