[Supplementary Material]

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

(a) $R_{\mathcal{C}} = 32$    (b) $R_{\mathcal{C}} = 256$    (c) $R_{\mathcal{C}} = 1024$

(d) $R_{\mathcal{C}} = 32$    (e) $R_{\mathcal{C}} = 256$    (f) $R_{\mathcal{C}} = 1024$

Figure 1: Testing accuracy with different algorithms, for WideResNet-40-8 on CIFAR-100.

In Figure 1 and Figure 2, we show the test accuracy on CIFAR-100 and ImageNet respectively, with the overall compression ratios in $\{32, 256, 1024\}$. Since the experiments on ImageNet are expensive, we do not tune different configurations of compressors ($H, R_{\mathcal{C}_1}, R_{\mathcal{C}_2}$) for each overall $R_{\mathcal{C}}$ on ImageNet, but directly use the best configurations tuned on CIFAR-100.

### 5.3 Discussion

We can see that in all the experiments, with the same compression ratio, CSER shows better performance than the baselines. When the compression ratio is small enough ($\leq 16$), the test accuracy is even better than fully synchronous SGD on CIFAR-100. When $R_{\mathcal{C}} \leq 32$, for CIFAR-100, QSparse-local-SGD has comparable performance to CSER or its special cases. Even with very large

Figure 2: Testing accuracy with different algorithms, for ResNet-50 on ImageNet.

compression ratio ($\approx 256$), the proposed algorithm can achieve comparable accuracy to SGD with full precision. CSER accelerates training by $10\times$ for CIFAR-100, and $4.5\times$ for ImageNet.

Compared to EF-SGD, CSER shows much better performance when the compression ratio is large ($R_{\mathcal{C}} \geq 64$). We can see that CSER fixes the convergence issue in EF-SGD and QSparse-local-SGD when aggressive compressors are used, as discussed in Section 3.1. For ImageNet with $R_C = 1024$, even if we decrease the learning rates to $0.025$, EF-SGD and QSparse-local-SGD still diverge, while CSER still converges well with even larger learning rates.

Note that in most cases, CSER performs better than CSEA and CSER-PL. The reason is that CSER uses both gradient partial synchronization and model partial synchronization. With finely tuned compression ratios, the local models will not be too far away from each other between the model synchronization rounds, which results in better convergence. Note that although CSEA has slightly worse performance compared to CSER and CSER-PL, it has less hyperparameters to be tuned for the compressors, which is more user-friendly in practice.

## 6  Conclusion

We proposed a novel communication-efficient SGD algorithm called CSER. We introduce error reset and partial synchronization that enable an aggressive compression rate as high as $1024\times$. Theoretically, we show that the proposed algorithm enjoys a linear speedup using more workers. Our empirical results show that the proposed algorithm accelerates the training of deep neural networks. In future work, we will apply our algorithms to other datasets and applications.

## Broader Impact

As this work is mainly algorithmic, the impact is mainly in scientific aspects rather than ethical and societal aspects. Hopefully, our work would enable faster training of machine learning models without regression in accuracy. It would save not only the time but also the expense cost by training large and complex models. On the other hand, there are some related aspects that we have not studied in this work. For example, we do not know how our approaches impact fairness and privacy of the model training, which will be our future work.

## Acknowledgments and Disclosure of Funding

This work was funded in part by the following grants: NSF IIS 1909577, NSF CNS 1908888, NSF CCF 1934986 and a JP Morgan Chase Fellowship, along with computational resources donated by Intel, AWS, and Microsoft Azure.

## Competing Interests

Part-time employment with Google Research Accra (OK).

## Footnotes

*The work was done when Cong Xie was a (part-time) intern in Amazon Web Services.

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

# Appendix

## A    Special cases

In this section, we introduce some important special cases of CSER, as well as the memory-efficient implementation of CSER with GRBS as the compressor. The corresponding experiments are shown in Appendix D.

### A.1    Special cases of CSER

By specifying $\delta_1$, $\delta_2$ and $H$, we recover some important special cases of CSER. Some existing approaches are similar to these special cases, though the differences often turn out to be important.

#### A.1.1    SGD with error assimilation

Taking $\mathcal{C}_2(v) = 0$ and $H = 1$, we recover a special case similar to EF-SGD [9]. However, different from EF-SGD, our special case directly assimilates the remaining error into the local model used for gradient computation in the next iteration. We name this special case "communication-efficient SGD with error assimilation" (CSEA). Importantly, "error assimilation" results in bifurcated local models without staleness in the local residuals– in contrast, "error feedback" of EF-SGD always produces synchronized local models but delayed local residuals. Thus, CSEA trades off the synchronization of the local models for the elimination of the staleness in the local residuals, and potentially mitigates the noise caused by staleness when high compression ratios are used.

---

**Algorithm 5** CSER

1: **Input**: $\mathcal{C}_1, \mathcal{C}_2$ - compressors, $H > 0$ - synchronization interval
2: Initialize $x_{i,0} = \hat{x}_0 \in \mathbb{R}^d, e_{i,0} = \mathbf{0}, \forall i \in [n]$
3: **for all** iteration $t \in [T]$ **do**
4:     **for all** Workers $i \in [n]$ in parallel **do**
5:         $g_{i,t} \leftarrow \nabla f(x_{i,t-1}; z_{i,t}), z_{i,t} \sim \mathcal{D}_i$
6:         $g'_{i,t}, r_{i,t} \leftarrow PSync(g_{i,t}, \mathcal{C}_2)$
7:         $x_{i,t-\frac{1}{2}} \leftarrow x_{i,t-1} - \eta g'_{i,t}$
8:         $e_{i,t-\frac{1}{2}} \leftarrow e_{i,t-1} - \eta r_{i,t}$
9:         **if** $\mod(t, H) \neq 0$ **then**
10:             $x_{i,t} \leftarrow x_{i,t-\frac{1}{2}}, \quad e_{i,t} \leftarrow e_{i,t-\frac{1}{2}}$
11:         **else**
12:             $e'_{i,t-\frac{1}{2}}, e_{i,t} \leftarrow PSync(e_{i,t-\frac{1}{2}}, \mathcal{C}_1)$
13:             $x_{i,t} \leftarrow x_{i,t-\frac{1}{2}} - e_{i,t-\frac{1}{2}} + e'_{i,t-\frac{1}{2}}$
14:         **end if**
15:     **end for**
16: **end for**

---

**Algorithm 6** Partial Synchronization (PSync)

1: **function** PSYNC($v_i \in \mathbb{R}^d, \mathcal{C}$ - compressor)
2:     On worker $i$:
3:     $v'_i = \mathcal{C}(v_i)$
4:     $r_i = v_i - v'_i$
5:     Partial synchronization:
6:     $\bar{v}' = \frac{1}{n} \sum_{i \in [n]} v'_i$
7:     $v'_i = \bar{v}' + r_i$
8:     **return** $v'_i, r_i$
9: **end function**

---

**Algorithm 7** CSEA (Implementation I)

1: **Input**: $\mathcal{C}_1$ - compressor
2: Initialize $x_{i,0} = \hat{x}_0 \in \mathbb{R}^d, e_{i,0} = \mathbf{0}, \forall i \in [n]$
3: **for all** iteration $t \in [T]$ **do**
4:     **for all** Workers $i \in [n]$ in parallel **do**
5:         $p_{i,t} \leftarrow e_{i,t-1} - \eta \nabla f(x_{i,t-1}; z_{i,t})$
6:         $e'_{i,t}, e_{i,t} \leftarrow PSync(p_{i,t}, \mathcal{C}_1)$
7:         $x_{i,t} \leftarrow x_{i,t-1} + e'_{i,t} - e_{i,t-1}$
8:     **end for**
9: **end for**

---

**Algorithm 8** CSER-PL (Implementation I)

1: **Input**: $\mathcal{C}_1$ - compressor, $H > 0$ - synchronization interval
2: Initialize $x_{i,0} = \hat{x}_0 \in \mathbb{R}^d, e_{i,0} = \mathbf{0}, \forall i \in [n]$
3: **for all** iteration $t \in [T]$ **do**
4:     **for all** Workers $i \in [n]$ in parallel **do**
5:         $g_{i,t} \leftarrow \nabla f(x_{i,t-1}; z_{i,t})$
6:         $x_{i,t-\frac{1}{2}} \leftarrow x_{i,t-1} - \eta g_{i,t}$
7:         $e_{i,t-\frac{1}{2}} \leftarrow e_{i,t-1} - \eta g_{i,t}$
8:         **if** $\mod(t, H) \neq 0$ **then**
9:             $x_{i,t} \leftarrow x_{i,t-\frac{1}{2}}, \quad e_{i,t} \leftarrow e_{i,t-\frac{1}{2}}$
10:         **else**
11:             $e'_{i,t}, e_{i,t} \leftarrow PSync(e_{i,t-\frac{1}{2}}, \mathcal{C}_1)$
12:             $x_{i,t} \leftarrow x_{i,t-\frac{1}{2}} + e'_{i,t} - e_{i,t-\frac{1}{2}}$
13:         **end if**
14:     **end for**
15: **end for**

### A.1.2 Partial-local-SGD

Taking $\mathcal{C}_2(v) = 0$, we recover a special case of CSER, which is similar to QSparse-local-SGD [3]. To distinguish it from Qsparse-local-SGD, we call this special case "Partial-local-SGD", or CSER-PL (partial-local special case of CSER) in brief. While QSparse-local-SGD keeps the local models fully synchronized after every communication round, CSER-PL maintains different local models across the workers. Taking $\delta_1 = 1$, CSER-PL recovers local SGD with synchronization interval $H$.

## A.2 Comparision to the existing work

### A.2.1 CSEA

If we take $\mathcal{C}_2(v) = 0$ (i.e., no synchronization at all) and $H = 1$, then we recover a special case similar to EF-SGD, with the same communication overhead, if the same compressor $\mathcal{C}_1$ is used.

| **Algorithm 9** CSEA | **Algorithm 10** EF-SGD |
|---|---|
| 1: **Input**: $\mathcal{C}_1$ - compressor | 1: **Input**: $\mathcal{C}_1$ - compressor |
| 2: Initialize $x_{i,0} = \hat{x}_0 \in \mathbb{R}^d, e_{i,t} = \mathbf{0}, \forall i \in [n]$ | 2: Initialize $x_{i,0} = \hat{x}_0 \in \mathbb{R}^d, e_{i,t} = \mathbf{0}, \forall i \in [n]$ |
| 3: **for all** iteration $t \in [T]$ **do** | 3: **for all** iteration $t \in [T]$ **do** |
| 4:     **for all** Workers $i \in [n]$ in parallel **do** | 4:     **for all** Workers $i \in [n]$ in parallel **do** |
| 5:         $p_{i,t} \leftarrow e_{i,t-1} - \eta \nabla f(x_{i,t-1}; z_{i,t})$ | 5:         $p_{i,t} \leftarrow e_{i,t-1} - \eta \nabla f(x_{i,t-1}; z_{i,t})$ |
| 6:         $p'_{i,t} \leftarrow \mathcal{C}_1(p_{i,t})$ | 6:         $p'_{i,t} \leftarrow \mathcal{C}_1(p_{i,t})$ |
| 7:         $e_{i,t} \leftarrow p_{i,t} - p'_{i,t}$ | 7:         $e_{i,t} \leftarrow p_{i,t} - p'_{i,t}$ |
| 8:         $\bar{p}'_t \leftarrow \frac{1}{n} \sum_{i \in [n]} p'_{i,t}$ | 8:         $\bar{p}'_t \leftarrow \frac{1}{n} \sum_{i \in [n]} p'_{i,t}$ |
| 9:         $\boxed{x_{i,t} \leftarrow x_{i,t-1} - e_{i,t-1} + e_{i,t} + \bar{p}'_t}$ | 9:         $\boxed{x_{i,t} \leftarrow x_{i,t-1} + \bar{p}'_t}$ |
| 10:     **end for** | 10:     **end for** |
| 11: **end for** | 11: **end for** |

## A.3 CSER-PL

By taking $\mathcal{C}_2(v) = 0$ (i.e., no partial synchronization of gradients), we recover a special case of CSER, which is similar to Qspars-local-SGD. To distinguish it from Qsparse-local-SGD, we call this special case "CSER-PL", or CSER-PL in brief. The major differences between CSER-PL and Qspars-local-SGD are shown below.

Note that we rewrite both algorithms into a new format for an easier comparison. Some notations are inconsistent to Algorithm 2.

CSER-PL is more memory-efficient compared to Qsparse-local-SGD, since it does not maintain the variable $r_{i,t}$ during the local updates.

| **Algorithm 11** CSER-PL | **Algorithm 12** Qsparse-local-SGD |
|---|---|
| 1: **Input**: $\mathcal{C}_1$ - compressor, $H > 0$ - synchronization interval | 1: **Input**: $\mathcal{C}_1$ - compressor, $H > 0$ - synchronization interval |
| 2: Initialize $x_{i,0} = \hat{x}_0 = \mathbf{0}, \forall i \in [n]$ | 2: Initialize $x_{i,0} = \hat{x}_0 = \mathbf{0}, \forall i \in [n]$ |
| 3: Pass | 3: Initialize $e_{i,0} = \mathbf{0}, \forall i \in [n]$ |
| 4: **for all** iteration $t \in [T]$ **do** | 4: **for all** iteration $t \in [T]$ **do** |
| 5:     **for all** Workers $i \in [n]$ in parallel **do** | 5:     **for all** Workers $i \in [n]$ in parallel **do** |
| 6:         $x_{i,t-\frac{1}{2}} \leftarrow x_{i,t-1} - \eta \nabla f(x_{i,t-1}; z_{i,t})$ | 6:         $x_{i,t-\frac{1}{2}} \leftarrow x_{i,t-1} - \eta \nabla f(x_{i,t-1}; z_{i,t})$ |
| 7:         **if** $\mod (t, H) \neq 0$ **then** | 7:         **if** $\mod (t, H) \neq 0$ **then** |
| 8:           $x_{i,t} \leftarrow x_{i,t-\frac{1}{2}}$ | 8:           $x_{i,t} \leftarrow x_{i,t-\frac{1}{2}}$ |
| 9:           $\hat{x}_t \leftarrow \hat{x}_{t-1}$ | 9:           $\hat{x}_t \leftarrow \hat{x}_{t-1}$ |
| 10:           Pass | 10:           $e_{i,t} \leftarrow e_{i,t-1}$ |
| 11:         **else** | 11:         **else** |
| 12:           $p_{i,t} \leftarrow x_{i,t-\frac{1}{2}} - \hat{x}_t$ | 12:           $p_{i,t} \leftarrow e_{i,t-1} + x_{i,t-\frac{1}{2}} - \hat{x}_{t-1}$ |
| 13:           $p'_{i,t} \leftarrow \mathcal{C}_1(p_{i,t})$ | 13:           $p'_{i,t} \leftarrow \mathcal{C}_1(p_{i,t})$ |
| 14:           $e_{i,t} \leftarrow p_{i,t} - p'_{i,t}$ | 14:           $e_{i,t} \leftarrow p_{i,t} - p'_{i,t}$ |
| 15:           $\bar{p}'_t \leftarrow \frac{1}{n} \sum_{i \in [n]} p'_{i,t}$ | 15:           $\bar{p}'_t \leftarrow \frac{1}{n} \sum_{i \in [n]} p'_{i,t}$ |
| 16:           $x_{i,t} \leftarrow \hat{x}_{t-1} + \bar{p}'_t + e_{i,t}$ | 16:           $x_{i,t} \leftarrow \hat{x}_{t-1} + \bar{p}'_t$ |
| 17:           $\hat{x}_t \leftarrow \hat{x}_{t-1} + \bar{p}'_t$ | 17:           $\hat{x}_t \leftarrow \hat{x}_{t-1} + \bar{p}'_t$ |
| 18:         **end if** | 18:         **end if** |
| 19:     **end for** | 19:     **end for** |
| 20: **end for** | 20: **end for** |

## A.4 Special implementations with GRBS

Using GRBS as the compressor, the implementation of CSER can be simplified. For any block, its local residual is either already assimilated into the local model, or reset to 0. Thus, we can directly do partial synchronization on the local models $x_{i,t}$, instead of the residuals $e_{i,t}$. The detailed implementations are shown in Algorithm 13, 14, and 15.

| **Algorithm 13** CSER (implementation II) | **Algorithm 14** CSER-PL (implementation II) |
|---|---|
| 1: **Input**: $\mathcal{C}_1, \mathcal{C}_2$ - randomized sparsifiers, $H > 0$ - synchronization interval | 1: **Input**: $\mathcal{C}_1$ - randomized sparsifier, $H > 0$ - synchronization interval |
| 2: $x_{i,0} = \hat{x}_0 \in \mathbb{R}^d, m_{i,0} = \mathbf{0}, \forall i \in [n]$ | 2: $x_{i,0} = \hat{x}_0 \in \mathbb{R}^d, m_{i,0} = \mathbf{0}, \forall i \in [n]$ |
| 3: **for all** iteration $t \in [T]$ **do** | 3: **for all** iteration $t \in [T]$ **do** |
| 4:     **for all** Workers $i \in [n]$ in parallel **do** | 4:     **for all** Workers $i \in [n]$ in parallel **do** |
| 5:         $p_{i,t} \leftarrow Get\_p(x_{i,t-1}, m_{i,t-1})$ | 5:         $p_{i,t} \leftarrow Get\_p(x_{i,t-1}, m_{i,t-1})$ |
| 6:         $p'_{i,t}, \_ \leftarrow PSync(p_{i,t}, \mathcal{C}_2)$ | 6:         $x_{i,t-\frac{1}{2}} \leftarrow x_{i,t-1} - p_{i,t}$ |
| 7:         $x_{i,t-\frac{1}{2}} \leftarrow x_{i,t-1} - p'_{i,t}$ | 7:         **if** $\mod (t, H) \neq 0$ **then** |
| 8:         **if** $\mod (t, H) \neq 0$ **then** | 8:           $x_{i,t} \leftarrow x_{i,t-\frac{1}{2}}$ |
| 9:           $x_{i,t} \leftarrow x_{i,t-\frac{1}{2}}$ | 9:         **else** |
| 10:         **else** | 10:           $x_{i,t}, \_ \leftarrow PSync(x_{i,t-\frac{1}{2}}, \mathcal{C}_1)$ |
| 11:           $x_{i,t}, \_ \leftarrow PSync(x_{i,t-\frac{1}{2}}, \mathcal{C}_1)$ | 11:         **end if** |
| 12:         **end if** | 12:     **end for** |
| 13:     **end for** | 13: **end for** |
| 14: **end for** | |

| **Algorithm 15** CSEA (implementation II) | **Algorithm 16** Calculate update on worker $i$ |
|---|---|
| 1: **Input**: $\mathcal{C}_1$ - randomized sparsifier | 1: **function** GET_P$(x_{i,t-1}, m_{i,t-1})$ |
| 2: $x_{i,0} = \hat{x}_0 \in \mathbb{R}^d, m_{i,0} = \mathbf{0}, \forall i \in [n]$ | 2: $\quad g_{i,t} \leftarrow \nabla f(x_{i,t-1}; z_{i,t}), z_{i,t} \sim \mathcal{D}_i$ |
| 3: **for all** iteration $t \in [T]$ **do** | 3: $\quad$ **if** use momentum **then** |
| 4: $\quad$ **for all** Workers $i \in [n]$ in parallel **do** | 4: $\qquad m_{i,t} \leftarrow \beta m_{i,t-1} + g_{i,t}$ |
| 5: $\qquad p_{i,t} \leftarrow Get\_p(x_{i,t-1}, m_{i,t-1})$ | 5: $\qquad p_{i,t} \leftarrow \eta(\beta m_{i,t} + g_{i,t})$ |
| 6: $\qquad x_{i,t-\frac{1}{2}} \leftarrow x_{i,t-1} - p_{i,t}$ | 6: $\quad$ **else** |
| 7: $\qquad x_{i,t,\_} \leftarrow PSync(x_{i,t-\frac{1}{2}}, \mathcal{C}_1)$ | 7: $\qquad p_{i,t} \leftarrow \eta g_{i,t}$ |
| 8: $\quad$ **end for** | 8: $\quad$ **end if** |
| 9: **end for** | 9: $\quad$ **return** $p_{i,t}$ |
|  | 10: **end function** |

# B  Proofs

**Lemma 1.** *(Bifurcated local models)* $e_{i,t}, \forall i \in [n], t$ *maintains the differences between the local models* $x_{i,t}$:

$$x_{i,t} - e_{i,t} = x_{j,t} - e_{j,t}, \forall i, j \in [n], t.$$

*Proof.* We prove the lemma by induction.

For $t = 0$, we have $x_{i,0} = \hat{x}_0, e_{i,0} = \mathbf{0}, \forall i \in [n]$, thus $x_{i,0} - e_{i,0} = x_{j,0} - e_{j,0}, \forall i, j \in [n]$.

Assume that

$$x_{i,t-1} - e_{i,t-1} = x_{j,t-1} - e_{j,t-1}, \forall i, j \in [n],$$

then we have 2 cases:

**Case 1:** $\mod(t, H) \neq 0$. Then we have

$$x_{i,t} = x_{i,t-\frac{1}{2}} = x_{i,t-1} - \eta \left[ \frac{1}{n} \sum_{k \in [n]} \mathcal{C}_2(g_{k,t}) + r_{i,t} \right],$$

$$e_{i,t} = e_{i,t-\frac{1}{2}} = e_{i,t-1} - \eta r_{i,t}.$$

Thus, we have

$$x_{i,t} - e_{i,t} = x_{i,t-1} - e_{i,t-1} - \eta \frac{1}{n} \sum_{k \in [n]} \mathcal{C}_2(g_{k,t}) = x_{j,t-1} - e_{j,t-1} - \eta \frac{1}{n} \sum_{k \in [n]} \mathcal{C}_2(g_{k,t}) = x_{j,t} - e_{j,t}$$

**Case 2:** $\mod(t, H) = 0$. Note that in Case 1 we have already proved

$$x_{i,t-\frac{1}{2}} - e_{i,t-\frac{1}{2}} = x_{j,t-\frac{1}{2}} - e_{j,t-\frac{1}{2}}, \forall i, j \in [n].$$

Then, we have

$$x_{i,t} = x_{i,t-\frac{1}{2}} - e_{i,t-\frac{1}{2}} + e'_{i,t-\frac{1}{2}} = x_{i,t-\frac{1}{2}} - e_{i,t-\frac{1}{2}} + \left[ \frac{1}{n} \sum_k \mathcal{C}_1(e_{k,t-\frac{1}{2}}) + e_{i,t} \right].$$

Thus, we get

$$x_{i,t} - e_{i,t} = x_{i,t-\frac{1}{2}} - e_{i,t-\frac{1}{2}} + \frac{1}{n} \sum_k \mathcal{C}_1(e_{k,t-\frac{1}{2}}) = x_{j,t-\frac{1}{2}} - e_{j,t-\frac{1}{2}} + \frac{1}{n} \sum_k \mathcal{C}_1(e_{k,t-\frac{1}{2}}) = x_{j,t} - e_{j,t}.$$

$\square$

**Lemma 3.** *(Error Reset of CSER) After every $H$ steps, the local error will be reset to*

$$\mathbb{E} \|e_{i,t}\|^2 \leq \frac{(1 - \delta_2)(1 - \delta_1)\eta^2 H^2 V_2}{\left(1 - \sqrt{1 - \delta_1}\right)^2},$$

*for* $\forall t \in [T], \mod(t, H) = 0$.

*Proof.* First, we establish the bound of the local error $\mathbb{E}\|e_{i,t-\frac{1}{2}}\|^2$ before (partial) synchronization.

**Case I:** For $t = 0$, we have no local error:

$$e_{i,0} = \mathbf{0}.$$

**Case II:** For $t = H$, we have the local error:

$$\mathbb{E}\|e_{i,t-\frac{1}{2}}\|^2$$

$$= \mathbb{E}\left\|-\eta \sum_{\tau=1}^{H} r_{i,\tau}\right\|^2$$

$$= \eta^2 \mathbb{E}\left\|\sum_{\tau=1}^{H} r_{i,\tau}\right\|^2$$

$$\leq \eta^2 H \sum_{\tau=1}^{H} \mathbb{E}\|r_{i,\tau}\|^2$$

$$= \eta^2 H \sum_{\tau=1}^{H} \mathbb{E}\|g_{i,\tau} - \mathcal{C}_1(g_{i,\tau})\|^2$$

$$\leq \eta^2 H(1-\delta_2) \sum_{\tau=1}^{H} \mathbb{E}\|g_{i,\tau}\|^2$$

$$\leq (1-\delta_2)\eta^2 H^2 V_2.$$

**Case III:** For any $t \in [T]$ such that $\mod(t,H) = 0, t > H$, we can bound the local error:

$$\mathbb{E}\|e_{i,t-\frac{1}{2}}\|^2 = \mathbb{E}\left\|e_{i,t-H} - \eta \sum_{\tau=t-H+1}^{t} r_{i,\tau}\right\|^2.$$

Note that after (partial) synchronization, the local error is reset as $e_{i,t-H} = e_{i,t-H-\frac{1}{2}} - \mathcal{C}_2\left(e_{i,t-H-\frac{1}{2}}\right)$.

Thus, we have

$$\mathbb{E}\|e_{i,t-\frac{1}{2}}\|^2$$

$$= \mathbb{E}\left\|e_{i,t-H-\frac{1}{2}} - \mathcal{C}_2\left(e_{i,t-H-\frac{1}{2}}\right) - \eta \sum_{\tau=t-H+1}^{t} (g_{i,\tau} - \mathcal{C}_1(g_{i,\tau}))\right\|^2$$

$$\leq (1+a)\mathbb{E}\left\|e_{i,t-H-\frac{1}{2}} - \mathcal{C}_2\left(e_{i,t-H-\frac{1}{2}}\right)\right\|^2 + (1+1/a)\mathbb{E}\left\|\eta \sum_{\tau=t-H+1}^{t} (g_{i,\tau} - \mathcal{C}_1(g_{i,\tau}))\right\|^2$$

$$\leq (1+a)(1-\delta_1)\mathbb{E}\left\|e_{i,t-H-\frac{1}{2}}\right\|^2 + (1+1/a)\eta^2 H(1-\delta_2) \sum_{\tau=t-H+1}^{t} \mathbb{E}\|g_{i,\tau}\|^2$$

$$\leq (1+a)(1-\delta_1)\mathbb{E}\left\|e_{i,t-H-\frac{1}{2}}\right\|^2 + (1+1/a)(1-\delta_2)\eta^2 H^2 V_2$$

$$\leq (1+1/a)(1-\delta_2)\eta^2 H^2 V_2 \sum_{\tau=0}^{+\infty} [(1+a)(1-\delta_1)]^\tau$$

$$\leq \frac{1+1/a}{1-(1+a)(1-\delta_1)}(1-\delta_2)\eta^2 H^2 V_2,$$

for any $a > 0$, such that $(1+a)(1-\delta_1) \in (0,1)$. The bound above is minimized when we take $a = \frac{1}{\sqrt{1-\delta_1}} - 1$, which results in

$$\mathbb{E}\|e_{i,t-\frac{1}{2}}\|^2 \leq \frac{(1-\delta_2)\eta^2 H^2 V_2}{\left(1-\sqrt{1-\delta_1}\right)^2}.$$

Combining all the 3 cases above, we obtain that for $\forall t \in [T]$ such that $\mod (t, H) = 0$:

$$\mathbb{E}\|e_{i,t-\frac{1}{2}}\|^2 \leq \frac{(1 - \delta_2)\eta^2 H^2 V_2}{\left(1 - \sqrt{1 - \delta_1}\right)^2}.$$

Then, after the (partial) synchronization, we have

$$\mathbb{E}\left\|e_{i,t}\right\|^2$$
$$\leq (1 - \delta_1)\mathbb{E}\left\|e_{i,t}\right\|^2$$
$$\leq \frac{(1 - \delta_2)(1 - \delta_1)\eta^2 H^2 V_2}{\left(1 - \sqrt{1 - \delta_1}\right)^2},$$

for $\forall t \in [T], \mod (t, H) = 0$. □

**Theorem 1.** *Taking $\eta \leq \frac{1}{L}$, after $T$ iterations, Algorithm 2 has the following error bound:*

$$\frac{1}{T} \sum_{t=1}^{T} \mathbb{E}\left[\|\nabla F(\bar{x}_{t-1})\|^2\right]$$

$$\leq \frac{2\left[F(\bar{x}_0) - F(x_*)\right]}{\eta T} + \left[\frac{(1 - \delta_1)}{\left(1 - \sqrt{1 - \delta_1}\right)^2} + 1\right] 2(1 - \delta_2)\eta^2 L^2 H^2 V_2 + \frac{L\eta V_1}{n}$$

$$\leq \frac{2\left[F(\bar{x}_0) - F(x_*)\right]}{\eta T} + \left[\frac{4(1 - \delta_1)}{\delta_1^2} + 1\right] 2(1 - \delta_2)\eta^2 L^2 H^2 V_2 + \frac{L\eta V_1}{n}.$$

*Proof.* Conditional on the previous states $(x_{i,t})$, using smoothness, we have

$$\mathbb{E}\left[F(\bar{x}_t)\right] \leq F(\bar{x}_{t-1}) + \underbrace{\mathbb{E}\left[\langle \nabla F(\bar{x}_{t-1}), \bar{x}_t - \bar{x}_{t-1}\rangle\right]}_{\textcircled{1}} + \frac{L}{2}\underbrace{\mathbb{E}\left[\|\bar{x}_t - \bar{x}_{t-1}\|^2\right]}_{\textcircled{2}}.$$

We bound the terms step by step.

Note that $\mathbb{E}\left[g_{i,t}\right] = \nabla F_i(x_{i,t-1})$. Thus, we have

$$\textcircled{2}$$
$$= \mathbb{E}\|\bar{x}_t - \bar{x}_{t-1}\|^2$$
$$= \mathbb{E}\left\|\frac{\eta}{n}\sum_{i\in[n]} g_{i,t}\right\|^2$$
$$= \eta^2 \mathbb{E}\left\|\frac{1}{n}\sum_{i\in[n]} (g_{i,t} - \nabla F_i(x_{i,t-1}) + \nabla F_i(x_{i,t-1}))\right\|^2$$
$$= \eta^2 \mathbb{E}\left\|\frac{1}{n}\sum_{i\in[n]} (g_{i,t} - \nabla F_i(x_{i,t-1}))\right\|^2 + \eta^2 \mathbb{E}\left\|\frac{1}{n}\sum_{i\in[n]} \nabla F_i(x_{i,t-1})\right\|^2$$
$$\leq \frac{\eta^2}{n} V_1 + \eta^2 \mathbb{E}\left\|\frac{1}{n}\sum_{i\in[n]} \nabla F_i(x_{i,t-1})\right\|^2.$$

$$\textcircled{1}$$
$$= \mathbb{E}\left[\langle \nabla F(\bar{x}_{t-1}), \bar{x}_t - \bar{x}_{t-1}\rangle\right]$$
$$= -\eta\mathbb{E}\left[\left\langle \nabla F(\bar{x}_{t-1}), \frac{1}{n}\sum_{i\in[n]} g_{i,t}\right\rangle\right]$$

$$= -\eta \left\langle \nabla F(\bar{x}_{t-1}), \frac{1}{n} \sum_{i \in [n]} \nabla F_i(x_{i,t-1}) \right\rangle$$

$$= -\frac{\eta}{2} \|\nabla F(\bar{x}_{t-1})\|^2 - \frac{\eta}{2} \left\| \frac{1}{n} \sum_{i \in [n]} \nabla F_i(x_{i,t-1}) \right\|^2 + \frac{\eta}{2} \underbrace{\left\| \nabla F(\bar{x}_{t-1}) - \frac{1}{n} \sum_{i \in [n]} \nabla F_i(x_{i,t-1}) \right\|^2}_{\text{③}}.$$

Using smoothness, we have

$$\text{③}$$

$$= \left\| \nabla F(\bar{x}_{t-1}) - \frac{1}{n} \sum_{i \in [n]} \nabla F_i(x_{i,t-1}) \right\|^2$$

$$= \left\| \frac{1}{n} \sum_{i \in [n]} \nabla F_i(\bar{x}_{t-1}) - \frac{1}{n} \sum_{i \in [n]} \nabla F_i(x_{i,t-1}) \right\|^2$$

$$\leq \frac{1}{n} \sum_{i \in [n]} \mathbb{E} \|\nabla F_i(\bar{x}_{t-1}) - \nabla F_i(x_{i,t-1})\|^2$$

$$\leq \frac{L^2}{n} \sum_{i \in [n]} \mathbb{E} \|\bar{x}_{t-1} - x_{i,t-1}\|^2.$$

Without loss of generality, assume that the latest synchronized model is $x_{i,t_0}$, where $t_0 \leq t-1$. Thus, we have

$$x_{i,t-1} = x_{i,t_0} - \eta \sum_{\tau=t_0+1}^{t-1} g'_{i,\tau},$$

$$\bar{x}_{t-1} = \frac{1}{n} \sum_{j \in [n]} x_{j,t_0} - \eta \sum_{\tau=t_0+1}^{t-1} \left( \frac{1}{n} \sum_{j \in [n]} g'_{i,\tau} \right),$$

which implies that

$$\bar{x}_{t-1} - x_{i,t-1} = \frac{1}{n} \sum_{j \in [n]} x_{j,t_0} - x_{i,t_0} + \eta \sum_{\tau=t_0+1}^{t-1} \left( g'_{i,\tau} - \frac{1}{n} \sum_{j \in [n]} g'_{j,\tau} \right).$$

It is easy to check that

$$\frac{1}{n} \sum_{j \in [n]} x_{j,t_0} - x_{i,t_0} = \frac{1}{n} \sum_{j \in [n]} e_{j,t_0} - e_{i,t_0},$$

and

$$g'_{i,\tau} - \frac{1}{n} \sum_{j \in [n]} g'_{j,\tau}$$

$$= r_{i,\tau} - \frac{1}{n} \sum_{j \in [n]} r_{j,\tau}$$

$$= g_{i,\tau} - \mathcal{C}_1 (g_{i,\tau}) - \frac{1}{n} \sum_{j \in [n]} (g_{j,\tau} - \mathcal{C}_1 (g_{i,\tau})).$$

Thus, we have

$$\text{③}$$

$$\leq \frac{L^2}{n} \sum_{i \in [n]} \mathbb{E} \left\| \frac{1}{n} \sum_{j \in [n]} x_{j,t_0} - x_{i,t_0} + \eta \sum_{\tau=t_0+1}^{t-1} \left( g'_{i,\tau} - \frac{1}{n} \sum_{j \in [n]} g'_{j,\tau} \right) \right\|^2$$

$$\leq \frac{2L^2}{n} \sum_{i \in [n]} \mathbb{E} \left\| \frac{1}{n} \sum_{j \in [n]} x_{j,t_0} - x_{i,t_0} \right\|^2 + \frac{2L^2}{n} \sum_{i \in [n]} \mathbb{E} \left\| \eta \sum_{\tau=t_0+1}^{t-1} \left( g'_{i,\tau} - \frac{1}{n} \sum_{j \in [n]} g'_{j,\tau} \right) \right\|^2$$

$$\leq \frac{2L^2}{n} \sum_{i \in [n]} \mathbb{E} \left\| \frac{1}{n} \sum_{j \in [n]} e_{j,t_0} - e_{i,t_0} \right\|^2$$

$$+ \frac{2L^2\eta^2 H}{n} \sum_{\tau=t_0+1}^{t-1} \sum_{i \in [n]} \mathbb{E} \left\| g_{i,\tau} - \mathcal{C}_1\left(g_{i,\tau}\right) - \frac{1}{n} \sum_{j \in [n]} \left( g_{j,\tau} - \mathcal{C}_1\left(g_{i,\tau}\right) \right) \right\|^2$$

$$\leq \frac{2L^2}{n} \sum_{i \in [n]} \mathbb{E} \left\| e_{i,t_0} \right\|^2 + \frac{2L^2\eta^2 H}{n} \sum_{\tau=t_0+1}^{t-1} \sum_{i \in [n]} \mathbb{E} \left\| g_{i,\tau} - \mathcal{C}_1\left(g_{i,\tau}\right) \right\|^2$$

$$\leq \frac{(1-\delta_2)(1-\delta_1)\eta^2 H^2 2 L^2 V_2}{\left(1 - \sqrt{1-\delta_1}\right)^2} + \frac{2L^2\eta^2 H(1-\delta_2)}{n} \sum_{\tau=t_0+1}^{t-1} \sum_{i \in [n]} \mathbb{E} \left\| g_{i,\tau} \right\|^2$$

$$\leq \frac{2(1-\delta_2)(1-\delta_1)\eta^2 H^2 L^2 V_2}{\left(1 - \sqrt{1-\delta_1}\right)^2} + 2(1-\delta_2)\eta^2 L^2 H^2 V_2$$

Put together all the ingredients above, using $\eta \leq \frac{1}{L}$, we have

$$\mathbb{E}\left[F(\bar{x}_t)\right]$$

$$\leq F(\bar{x}_{t-1}) + \textcircled{1} + \frac{L}{2}\textcircled{2}$$

$$\leq F(\bar{x}_{t-1}) - \frac{\eta}{2} \left\| \nabla F(\bar{x}_{t-1}) \right\|^2 - \frac{\eta}{2} \left\| \frac{1}{n} \sum_{i \in [n]} \nabla F_i(x_{i,t-1}) \right\|^2 + \frac{\eta}{2}\textcircled{3} + \frac{L}{2}\textcircled{2}$$

$$\leq F(\bar{x}_{t-1}) - \frac{\eta}{2} \left\| \nabla F(\bar{x}_{t-1}) \right\|^2 - \frac{\eta}{2} \left\| \frac{1}{n} \sum_{i \in [n]} \nabla F_i(x_{i,t-1}) \right\|^2$$

$$+ \frac{(1-\delta_2)(1-\delta_1)\eta^3 H^2 L^2 V_2}{\left(1 - \sqrt{1-\delta_1}\right)^2} + (1-\delta_2)\eta^3 L^2 H^2 V_2$$

$$+ \frac{L}{2} \left[ \frac{\eta^2}{n} V_1 + \eta^2 \mathbb{E} \left\| \frac{1}{n} \sum_{i \in [n]} \nabla F_i(x_{i,t-1}) \right\|^2 \right]$$

$$= F(\bar{x}_{t-1}) - \frac{\eta}{2} \left\| \nabla F(\bar{x}_{t-1}) \right\|^2 + \frac{L\eta^2 - \eta}{2} \left\| \frac{1}{n} \sum_{i \in [n]} \nabla F_i(x_{i,t-1}) \right\|^2$$

$$+ \frac{(1-\delta_2)(1-\delta_1)\eta^3 H^2 L^2 V_2}{\left(1 - \sqrt{1-\delta_1}\right)^2} + (1-\delta_2)\eta^3 L^2 H^2 V_2 + \frac{L\eta^2}{2n} V_1$$

$$= F(\bar{x}_{t-1}) - \frac{\eta}{2} \left\| \nabla F(\bar{x}_{t-1}) \right\|^2 + \left[ \frac{(1-\delta_1)}{\left(1 - \sqrt{1-\delta_1}\right)^2} + 1 \right] (1-\delta_2)\eta^3 L^2 H^2 V_2 + \frac{L\eta^2}{2n} V_1.$$

By telescoping and taking total expectation, after $T$ iterations, we have

$$\frac{1}{T}\sum_{t=1}^{T}\mathbb{E}\left[\|\nabla F(\bar{x}_{t-1})\|^2\right]$$

$$\leq \frac{2\left[F(\bar{x}_0)-F(x_*)\right]}{\eta T}+\left[\frac{(1-\delta_1)}{\left(1-\sqrt{1-\delta_1}\right)^2}+1\right]2(1-\delta_2)\eta^2 L^2 H^2 V_2+\frac{L\eta V_1}{n}$$

$$\leq \frac{2\left[F(\bar{x}_0)-F(x_*)\right]}{\eta T}+\left[\frac{(1-\delta_1)}{\left(1-\sqrt{1-\delta_1}\right)^2}+1\right]2(1-\delta_2)\eta^2 L^2 H^2 V_2+\frac{L\eta V_1}{n}.$$

$\square$

**Corollary 1.** *Taking* $\eta = \min\left\{\frac{\gamma}{\sqrt{T/n}+\left[4(1-\delta_1)/\delta_1^2+1\right]^{1/3}2^{1/3}(1-\delta_2)^{1/3}H^{2/3}T^{1/3}},\frac{1}{L}\right\}$ *for some* $\gamma > 0$, *after* $T \gg n$ *iterations, Algorithm 2 (CSER) converges to a critical point:* $\frac{1}{T}\sum_{t=1}^{T}\mathbb{E}\left[\|\nabla F(\bar{x}_{t-1})\|^2\right] \leq \frac{1}{\sqrt{nT}}\left[\frac{F_o}{\gamma}+\gamma LV_1\right]+\frac{\left[4(1-\delta_1)/\delta_1^2+1\right]^{1/3}2^{1/3}(1-\delta_2)^{1/3}H^{2/3}}{T^{2/3}}\left[\frac{F_o}{\gamma}+\gamma^2 L^2 V_2\right] \leq \mathcal{O}\left(\frac{1}{\sqrt{nT}}\right).$

*Proof.* From Theorem 1, we have

$$\frac{1}{T}\sum_{t=1}^{T}\mathbb{E}\left[\|\nabla F(\bar{x}_{t-1})\|^2\right]$$

$$\leq \frac{2\left[F(\bar{x}_0)-F(x_*)\right]}{\eta T}+\left[\frac{4(1-\delta_1)}{\delta_1^2}+1\right]2(1-\delta_2)\eta^2 L^2 H^2 V_2+\frac{L\eta V_1}{n}.$$

Denote $F_o = 2\left[F(\bar{x}_0)-F(x_*)\right]$, and $C = \left[\frac{4(1-\delta_1)}{\delta_1^2}+1\right]2(1-\delta_2)H^2$. Taking $\eta = \min\left\{\frac{\gamma}{\sqrt{T/n}+C^{1/3}T^{1/3}},\frac{1}{L}\right\}$, we have

$$\frac{1}{T}\sum_{t=1}^{T}\mathbb{E}\left[\|\nabla F(\bar{x}_{t-1})\|^2\right]$$

$$\leq \frac{F_o}{\eta T}+C\eta^2 L^2 V_2+\frac{L\eta V_1}{n}$$

$$\leq \frac{F_o}{T}\frac{\sqrt{T/n}+C^{1/3}T^{1/3}}{\gamma}+CL^2 V_2\frac{\gamma^2}{C^{2/3}T^{2/3}}+\frac{LV_1}{n}\frac{\gamma}{\sqrt{T/n}}$$

$$\leq \frac{F_o}{\gamma\sqrt{nT}}+\frac{F_o C^{1/3}}{\gamma T^{2/3}}+\frac{C^{1/3}\gamma^2 L^2 V_2}{T^{2/3}}+\frac{\gamma LV_1}{\sqrt{nT}}$$

$$\leq \frac{1}{\sqrt{nT}}\left[\frac{F_o}{\gamma}+\gamma LV_1\right]+\frac{C^{1/3}}{T^{2/3}}\left[\frac{F_o}{\gamma}+\gamma^2 L^2 V_2\right].$$

$\square$

**Lemma 4.** *(Bounded update) For any local update, we have* $\mathbb{E}\|p_{i,t}\|^2 \leq \frac{\eta^2}{(1-\beta)^2}V_2$, $\forall i,t$.

*Proof.* It is easy to check that

$$m_{i,t}=\sum_{\tau=1}^{t}\beta^{t-\tau}g_{i,\tau}.$$

Thus, taking $s_t = \sum_{\tau=0}^{t}\beta^{\tau}$, we have

$$\mathbb{E}\|p_{i,t}\|^2$$

$$= \eta^2 \mathbb{E} \left\| \beta m_{i,t} + g_{i,t} \right\|^2$$

$$= \eta^2 \mathbb{E} \left\| \left( \beta \sum_{\tau=1}^{t} \beta^{t-\tau} g_{i,\tau} \right) + g_{i,t} \right\|^2$$

$$= \eta^2 s_t^2 \mathbb{E} \left\| \left( \sum_{\tau=1}^{t} \frac{\beta^{t-\tau+1}}{s_t} g_{i,\tau} \right) + \frac{1}{s_t} g_{i,t} \right\|^2$$

$$\leq \eta^2 s_t^2 \left[ \left( \sum_{\tau=1}^{t} \frac{\beta^{t-\tau+1}}{s_t} \mathbb{E} \left\| g_{i,\tau} \right\|^2 \right) + \frac{1}{s_t} \mathbb{E} \left\| g_{i,t} \right\|^2 \right] \qquad \triangleright \text{ Jensen's inequality}$$

$$\leq \eta^2 s_t^2 \left[ \left( \sum_{\tau=1}^{t} \frac{\beta^{t-\tau+1}}{s_t} V_2 \right) + \frac{1}{s_t} V_2 \right] \qquad \triangleright \text{ Assumption 2}$$

$$= \eta^2 s_t \left[ \left( \sum_{\tau=1}^{t} \beta^{t-\tau+1} V_2 \right) + V_2 \right]$$

$$= \eta^2 s_t \left[ \left( \sum_{\tau=1}^{t} \beta^{t-\tau+1} V_2 \right) + V_2 \right]$$

$$= \eta^2 s_t \left( \sum_{\tau=1}^{t+1} \beta^{t-\tau+1} V_2 \right)$$

$$= \eta^2 s_t^2 V_2$$

$$\leq \frac{\eta^2}{(1-\beta)^2} V_2.$$

$\square$

**Lemma 5.** *(Error Reset of M-CSER) With momentum, after every $H$ steps, the local error will be reset to*

$$\mathbb{E} \left\| e_{i,t} \right\|^2 \leq \frac{(1-\delta_2)(1-\delta_1)\eta^2 H^2 V_2}{\left( 1 - \sqrt{1-\delta_1} \right)^2 (1-\beta)^2},$$

*for $\forall t \in [T], \mod (t, H) = 0$.*

*Proof.* First, we establish the bound of the local error $\mathbb{E}\|e_{i,t-\frac{1}{2}}\|^2$ before (partial) synchronization.

**Case I:** For $t = 0$, we have no local error:

$$e_{i,0} = \mathbf{0}.$$

**Case II:** For $t = H$, we have the local error:

$$\mathbb{E}\|e_{i,t-\frac{1}{2}}\|^2$$

$$= \mathbb{E} \left\| - \sum_{\tau=1}^{H} r_{i,\tau} \right\|^2$$

$$= \mathbb{E} \left\| \sum_{\tau=1}^{H} r_{i,\tau} \right\|^2$$

$$\leq H \sum_{\tau=1}^{H} \mathbb{E} \left\| r_{i,\tau} \right\|^2$$

$$= H \sum_{\tau=1}^{H} \mathbb{E} \left\| p_{i,\tau} - \mathcal{C}_1 \left( p_{i,\tau} \right) \right\|^2$$

$$\leq H(1-\delta_2)\sum_{\tau=1}^{H}\mathbb{E}\left\|p_{i,\tau}\right\|^2$$

$$\leq \frac{(1-\delta_2)\eta^2 H^2}{(1-\beta)^2}V_2. \qquad\qquad \triangleright \text{ Lemma 4}$$

**Case III:** For any $t \in [T]$ such that $\mod(t,H)=0, t > H$, we can bound the local error:

$$\mathbb{E}\|e_{i,t-\frac{1}{2}}\|^2 = \mathbb{E}\left\|e_{i,t-H}-\sum_{\tau=t-H+1}^{t}r_{i,\tau}\right\|^2.$$

Note that after (partial) synchronization, the local error is reset as $e_{i,t-H} = e_{i,t-H-\frac{1}{2}} - \mathcal{C}_2\left(e_{i,t-H-\frac{1}{2}}\right).$

Thus, we have

$$\mathbb{E}\|e_{i,t-\frac{1}{2}}\|^2$$

$$= \mathbb{E}\left\|e_{i,t-H-\frac{1}{2}}-\mathcal{C}_2\left(e_{i,t-H-\frac{1}{2}}\right)-\sum_{\tau=t-H+1}^{t}\left(p_{i,\tau}-\mathcal{C}_1\left(p_{i,\tau}\right)\right)\right\|^2$$

$$\leq (1+a)\mathbb{E}\left\|e_{i,t-H-\frac{1}{2}}-\mathcal{C}_2\left(e_{i,t-H-\frac{1}{2}}\right)\right\|^2+(1+1/a)\mathbb{E}\left\|\sum_{\tau=t-H+1}^{t}\left(p_{i,\tau}-\mathcal{C}_1\left(p_{i,\tau}\right)\right)\right\|^2$$

$$\leq (1+a)(1-\delta_1)\mathbb{E}\left\|e_{i,t-H-\frac{1}{2}}\right\|^2+(1+1/a)\eta^2 H(1-\delta_2)\sum_{\tau=t-H+1}^{t}\mathbb{E}\left\|p_{i,\tau}\right\|^2$$

$$\leq (1+a)(1-\delta_1)\mathbb{E}\left\|e_{i,t-H-\frac{1}{2}}\right\|^2+\frac{(1+1/a)(1-\delta_2)\eta^2 H^2}{(1-\beta)^2}V_2$$

$$\leq \frac{(1+1/a)(1-\delta_2)\eta^2 H^2}{(1-\beta)^2}V_2\sum_{\tau=0}^{+\infty}\left[(1+a)(1-\delta_1)\right]^{\tau}$$

$$\leq \frac{1+1/a}{1-(1+a)(1-\delta_1)}\frac{(1-\delta_2)\eta^2 H^2}{(1-\beta)^2}V_2,$$

for any $a > 0$, such that $(1+a)(1-\delta_1) \in (0,1)$. The bound above is minimized when we take $a = \frac{1}{\sqrt{1-\delta_1}}-1$, which results in

$$\mathbb{E}\|e_{i,t-\frac{1}{2}}\|^2 \leq \frac{(1-\delta_2)\eta^2 H^2 V_2}{\left(1-\sqrt{1-\delta_1}\right)^2(1-\beta)^2}.$$

Combining all the 3 cases above, we obtain that for $\forall t \in [T]$ such that $\mod(t,H)=0$:

$$\mathbb{E}\|e_{i,t-\frac{1}{2}}\|^2 \leq \frac{(1-\delta_2)\eta^2 H^2 V_2}{\left(1-\sqrt{1-\delta_1}\right)^2(1-\beta)^2}.$$

Then, after the (partial) synchronization, we have

$$\mathbb{E}\left\|e_{i,t}\right\|^2$$

$$= \mathbb{E}\left\|e_{i,t-\frac{1}{2}}-\mathcal{C}_2\left(e_{i,t-\frac{1}{2}}\right)\right\|^2$$

$$\leq (1-\delta_1)\mathbb{E}\left\|e_{i,t-\frac{1}{2}}\right\|^2$$

$$\leq \frac{(1-\delta_2)(1-\delta_1)\eta^2 H^2 V_2}{\left(1-\sqrt{1-\delta_1}\right)^2(1-\beta)^2},$$

for $\forall t \in [T]$, $\mod(t,H)=0$. $\qquad\qquad\square$

**Theorem 2.** *Taking $\eta \le \min\{\frac{1}{2}, \frac{1-\beta}{2L}\}$, after $T$ iterations, Algorithm 4 has the following error bound:*

$$\frac{1}{T}\sum_{t=1}^{T}\mathbb{E}\left[\|\nabla F(\bar{x}_{t-1})\|^2\right]$$

$$\le \frac{2(1-\beta)\left[F(\bar{x}_0) - F(x_*)\right]}{\eta T} + \frac{\eta^2\beta^4 L^2 V_2}{(1-\beta)^4} + \frac{\eta L V_1}{n(1-\beta)}$$

$$+ \left(\frac{1-\delta_1}{\left(1-\sqrt{1-\delta_1}\right)^2} + 1\right)\frac{2(1-\delta_2)\eta^2 H^2 L^2 V_2}{(1-\beta)^2}$$

$$\le \frac{2(1-\beta)\left[F(\bar{x}_0) - F(x_*)\right]}{\eta T} + \frac{\eta^2\beta^4 L^2 V_2}{(1-\beta)^4} + \frac{\eta L V_1}{n(1-\beta)}$$

$$+ \left(\frac{4(1-\delta_1)}{\delta_1^2} + 1\right)\frac{2(1-\delta_2)\eta^2 H^2 L^2 V_2}{(1-\beta)^2}.$$

*Proof.* To prove the convergence, we introduce 2 sequences of auxiliary variables: $\{\bar{x}_t, t \ge 0\}$ and $\{\bar{z}_t, t \ge 0\}$, where

$$\bar{x}_t = \frac{1}{n}\sum_{i\in[n]}x_{i,t}, \forall t \ge 0,$$

$$\bar{z}_t = \begin{cases} x_0, & t = 0, \\ \bar{x}_t - \frac{\eta\beta^2}{1-\beta}\frac{1}{n}\sum_{i\in[n]}m_{i,t}, & t \ge 1. \end{cases}$$

For $t = 0$, we have

$$\bar{z}_{t+1} - \bar{z}_t$$

$$= \bar{z}_1 - \bar{z}_0$$

$$= \bar{x}_1 - \frac{\eta\beta^2}{1-\beta}\frac{1}{n}\sum_{i\in[n]}m_{i,1} - \bar{x}_0$$

$$= -\frac{1}{n}\sum_{i\in[n]}p_{i,1} - \frac{\eta\beta^2}{1-\beta}\frac{1}{n}\sum_{i\in[n]}m_{i,1}$$

$$= -\eta\frac{1}{n}\sum_{i\in[n]}(\beta m_{i,1} + g_{i,1}) - \frac{\eta\beta^2}{1-\beta}\frac{1}{n}\sum_{i\in[n]}m_{i,1}$$

$$= -\eta\frac{1}{n}\sum_{i\in[n]}\left[(\beta + \frac{\beta^2}{1-\beta})m_{i,1} + g_{i,1}\right]$$

$$= -\eta\frac{1}{n}\sum_{i\in[n]}\left[\frac{\beta}{1-\beta}g_{i,1} + g_{i,1}\right]$$

$$= -\frac{\eta}{1-\beta}\frac{1}{n}\sum_{i\in[n]}g_{i,t+1}$$

For $t \ge 1$, we have

$$\bar{z}_{t+1} - \bar{z}_t$$

$$= \bar{x}_{t+1} - \frac{\eta\beta^2}{1-\beta}\frac{1}{n}\sum_{i\in[n]}m_{i,t+1} - \bar{x}_t + \frac{\eta\beta^2}{1-\beta}\frac{1}{n}\sum_{i\in[n]}m_{i,t}$$

$$= -\frac{1}{n}\sum_{i\in[n]}p_{i,t+1} - \frac{\eta\beta^2}{1-\beta}\frac{1}{n}\sum_{i\in[n]}m_{i,t+1} + \frac{\eta\beta^2}{1-\beta}\frac{1}{n}\sum_{i\in[n]}m_{i,t}$$

$$= -\eta \frac{1}{n} \sum_{i \in [n]} (\beta m_{i,t+1} + g_{i,t+1}) - \frac{\eta \beta^2}{1-\beta} \frac{1}{n} \sum_{i \in [n]} m_{i,t+1} + \frac{\eta \beta^2}{1-\beta} \frac{1}{n} \sum_{i \in [n]} m_{i,t}$$

$$= -\eta \frac{1}{n} \sum_{i \in [n]} g_{i,t+1} - \frac{\eta \beta}{1-\beta} \frac{1}{n} \sum_{i \in [n]} m_{i,t+1} + \frac{\eta \beta^2}{1-\beta} \frac{1}{n} \sum_{i \in [n]} m_{i,t}$$

$$= -\eta \frac{1}{n} \sum_{i \in [n]} g_{i,t+1} - \frac{\eta \beta}{1-\beta} \frac{1}{n} \sum_{i \in [n]} (\beta m_{i,t} + g_{i,t+1}) + \frac{\eta \beta^2}{1-\beta} \frac{1}{n} \sum_{i \in [n]} m_{i,t}$$

$$= -\frac{\eta}{1-\beta} \frac{1}{n} \sum_{i \in [n]} g_{i,t+1}$$

Thus, for $\forall t \geq 0$, we have

$$\bar{z}_{t+1} - \bar{z}_t = -\frac{\eta}{1-\beta} \frac{1}{n} \sum_{i \in [n]} g_{i,t+1}. \tag{1}$$

Conditional on all the states previous to $x_{i,t}$, using smoothness, we have

$$\mathbb{E}\left[F(\bar{z}_t)\right] \leq F(\bar{z}_{t-1}) + \underbrace{\mathbb{E}\left[\langle \nabla F(\bar{z}_{t-1}), \bar{z}_t - \bar{z}_{t-1}\rangle\right]}_{\textcircled{1}} + \frac{L}{2} \underbrace{\mathbb{E}\left[\|\bar{z}_t - \bar{z}_{t-1}\|^2\right]}_{\textcircled{2}}.$$

We bound the terms step by step.

Note that $\mathbb{E}\left[g_{i,t}\right] = \nabla F_i(x_{i,t-1})$. Thus, for $\textcircled{2}$, we have

$$\textcircled{2}$$
$$= \mathbb{E}\|\bar{z}_t - \bar{z}_{t-1}\|^2$$

$$= \mathbb{E}\left\|\frac{\eta}{1-\beta} \frac{1}{n} \sum_{i \in [n]} g_{i,t}\right\|^2$$

$$= \frac{\eta^2}{(1-\beta)^2} \mathbb{E}\left\|\frac{1}{n} \sum_{i \in [n]} (g_{i,t} - \nabla F_i(x_{i,t-1}) + \nabla F_i(x_{i,t-1}))\right\|^2$$

$$= \frac{\eta^2}{(1-\beta)^2} \mathbb{E}\left\|\frac{1}{n} \sum_{i \in [n]} (g_{i,t} - \nabla F_i(x_{i,t-1}))\right\|^2 + \frac{\eta^2}{(1-\beta)^2} \mathbb{E}\left\|\frac{1}{n} \sum_{i \in [n]} \nabla F_i(x_{i,t-1})\right\|^2$$

$$= \frac{\eta^2}{n(1-\beta)^2} V_1 + \frac{\eta^2}{(1-\beta)^2} \mathbb{E}\left\|\frac{1}{n} \sum_{i \in [n]} \nabla F_i(x_{i,t-1})\right\|^2.$$

Then, for $\textcircled{1}$, we have

$$\textcircled{1}$$
$$= \mathbb{E}\left[\langle \nabla F(\bar{z}_{t-1}), \bar{z}_t - \bar{z}_{t-1}\rangle\right]$$

$$= \mathbb{E}\left[\left\langle \nabla F(\bar{z}_{t-1}), -\frac{\eta}{1-\beta} \frac{1}{n} \sum_{i \in [n]} g_{i,t}\right\rangle\right]$$

$$= -\frac{\eta}{1-\beta} \mathbb{E}\left\langle \nabla F(\bar{z}_{t-1}), \frac{1}{n} \sum_{i \in [n]} \nabla F_i(x_{i,t-1})\right\rangle$$

$$= -\frac{\eta}{1-\beta}\mathbb{E}\left\langle \nabla F(\bar{z}_{t-1}) - \nabla F(\bar{x}_{t-1}), \frac{1}{n}\sum_{i\in[n]}\nabla F_i(x_{i,t-1})\right\rangle \underbrace{\qquad\qquad\qquad\qquad}_{\text{③}} - \frac{\eta}{1-\beta}\mathbb{E}\left\langle \nabla F(\bar{x}_{t-1}), \frac{1}{n}\sum_{i\in[n]}\nabla F_i(x_{i,t-1})\right\rangle \underbrace{\qquad\qquad\qquad\qquad}_{\text{④}}.$$

For ③, we have

$$\text{③}$$
$$= -\frac{\eta}{1-\beta}\mathbb{E}\left\langle \nabla F(\bar{z}_{t-1}) - \nabla F(\bar{x}_{t-1}), \frac{1}{n}\sum_{i\in[n]}\nabla F_i(x_{i,t-1})\right\rangle$$
$$= -\frac{1}{1-\beta}\mathbb{E}\left\langle \nabla F(\bar{z}_{t-1}) - \nabla F(\bar{x}_{t-1}), \frac{\eta}{n}\sum_{i\in[n]}\nabla F_i(x_{i,t-1})\right\rangle$$
$$\leq \frac{1}{2(1-\beta)}\underbrace{\mathbb{E}\left\|\nabla F(\bar{z}_{t-1}) - \nabla F(\bar{x}_{t-1})\right\|^2}_{\text{⑤}} + \frac{\eta^2}{2(1-\beta)}\mathbb{E}\left\|\frac{1}{n}\sum_{i\in[n]}\nabla F_i(x_{i,t-1})\right\|^2,$$

where (using smoothness)

$$\text{⑤}$$
$$= \mathbb{E}\left\|\nabla F(\bar{z}_{t-1}) - \nabla F(\bar{x}_{t-1})\right\|^2$$
$$\leq L^2\mathbb{E}\left\|\bar{z}_{t-1} - \bar{x}_{t-1}\right\|^2$$
$$= L^2\mathbb{E}\left\|\frac{\eta\beta^2}{1-\beta}\frac{1}{n}\sum_{i\in[n]}m_{i,t-1}\right\|^2$$
$$= \frac{L^2\eta^2\beta^4}{(1-\beta)^2}\left\|\frac{1}{n}\sum_{i\in[n]}m_{i,t-1}\right\|^2$$
$$\leq \frac{L^2\eta^2\beta^4}{(1-\beta)^4}V_2.$$

For ④, we have

$$\text{④}$$
$$= -\frac{\eta}{1-\beta}\mathbb{E}\left\langle \nabla F(\bar{x}_{t-1}), \frac{1}{n}\sum_{i\in[n]}\nabla F_i(x_{i,t-1})\right\rangle$$
$$= -\frac{\eta}{1-\beta}\left[\frac{1}{2}\mathbb{E}\left\|\nabla F(\bar{x}_{t-1})\right\|^2 + \frac{1}{2}\mathbb{E}\left\|\frac{1}{n}\sum_{i\in[n]}\nabla F_i(x_{i,t-1})\right\|^2 - \frac{1}{2}\mathbb{E}\left\|\nabla F(\bar{x}_{t-1}) - \frac{1}{n}\sum_{i\in[n]}\nabla F_i(x_{i,t-1})\right\|^2\right]$$
$$= -\frac{\eta}{2(1-\beta)}\mathbb{E}\left\|\nabla F(\bar{x}_{t-1})\right\|^2 - \frac{\eta}{2(1-\beta)}\mathbb{E}\left\|\frac{1}{n}\sum_{i\in[n]}\nabla F_i(x_{i,t-1})\right\|^2$$
$$+ \frac{\eta}{2(1-\beta)}\underbrace{\mathbb{E}\left\|\nabla F(\bar{x}_{t-1}) - \frac{1}{n}\sum_{i\in[n]}\nabla F_i(x_{i,t-1})\right\|^2}_{\text{⑥}},$$

where

$$\text{⑥}$$

$$= \mathbb{E}\left\|\nabla F(\bar{x}_{t-1}) - \frac{1}{n}\sum_{i\in[n]}\nabla F_i(x_{i,t-1})\right\|^2$$

$$= \mathbb{E}\left\|\frac{1}{n}\sum_{i\in[n]}\nabla F_i(\bar{x}_{t-1}) - \frac{1}{n}\sum_{i\in[n]}\nabla F_i(x_{i,t-1})\right\|^2$$

$$\leq \frac{1}{n}\sum_{i\in[n]}\mathbb{E}\left\|\nabla F_i(\bar{x}_{t-1}) - \nabla F_i(x_{i,t-1})\right\|^2$$

$$\leq \frac{L^2}{n}\sum_{i\in[n]}\mathbb{E}\left\|\bar{x}_{t-1} - x_{i,t-1}\right\|^2.$$

Without loss of generality, assume that the latest synchronized model is $x_{i,t_0}$, where $t_0 \leq t-1$, mod $(t_0, H) = 0$. Thus, we have

$$x_{i,t-1} = x_{i,t_0} - \sum_{\tau=t_0+1}^{t-1} p'_{i,\tau},$$

$$\bar{x}_{t-1} = \frac{1}{n}\sum_{j\in[n]} x_{j,t_0} - \sum_{\tau=t_0+1}^{t-1}\left(\frac{1}{n}\sum_{j\in[n]} p_{i,\tau}\right),$$

which implies that

$$\bar{x}_{t-1} - x_{i,t-1} = \frac{1}{n}\sum_{j\in[n]} x_{j,t_0} - x_{i,t_0} + \sum_{\tau=t_0+1}^{t-1}\left(p'_{i,\tau} - \frac{1}{n}\sum_{j\in[n]} p_{j,\tau}\right).$$

It is easy to check that

$$\frac{1}{n}\sum_{j\in[n]} x_{j,t_0} - x_{i,t_0}$$

$$= \frac{1}{n}\sum_{j\in[n]} e_{j,t_0} - e_{i,t_0},$$

and

$$p'_{i,\tau} - \frac{1}{n}\sum_{j\in[n]} p_{j,\tau}$$

$$= r_{i,\tau} - \frac{1}{n}\sum_{j\in[n]} r_{j,\tau}$$

$$= p_{i,\tau} - \mathcal{C}_1(p_{i,\tau}) - \frac{1}{n}\sum_{j\in[n]}(p_{j,\tau} - \mathcal{C}_1(p_{j,\tau}))$$

Then, we have

⑥

$$\leq \frac{L^2}{n}\sum_{i\in[n]}\mathbb{E}\left\|\bar{x}_{t-1} - x_{i,t-1}\right\|^2$$

$$\leq \frac{2L^2}{n}\sum_{i\in[n]}\mathbb{E}\left\|\frac{1}{n}\sum_{j\in[n]} e_{j,t_0} - e_{i,t_0}\right\|^2$$

$$+ \frac{2L^2}{n} \sum_{i\in[n]} \mathbb{E} \left\| \sum_{\tau=t_0+1}^{t-1} \left[ p_{i,\tau} - \mathcal{C}_1(p_{i,\tau}) - \frac{1}{n} \sum_{j\in[n]} (p_{j,\tau} - \mathcal{C}_1(p_{j,\tau})) \right] \right\|^2$$

$$\leq \frac{2L^2}{n} \sum_{i\in[n]} \mathbb{E} \|e_{i,t_0}\|^2 + \frac{2L^2 H}{n} \sum_{i\in[n]} \sum_{\tau=t_0+1}^{t-1} \mathbb{E} \|p_{i,\tau} - \mathcal{C}_1(p_{i,\tau})\|^2$$

$$\leq \frac{2(1-\delta_2)(1-\delta_1)\eta^2 H^2 L^2 V_2}{\left(1-\sqrt{1-\delta_1}\right)^2 (1-\beta)^2} + \frac{2L^2 H}{n} \sum_{i\in[n]} \sum_{\tau=t_0+1}^{t-1} \mathbb{E} \|p_{i,\tau} - \mathcal{C}_1(p_{i,\tau})\|^2 \qquad \triangleright \text{Lemma 5}$$

$$\leq \frac{2(1-\delta_2)(1-\delta_1)\eta^2 H^2 L^2 V_2}{\left(1-\sqrt{1-\delta_1}\right)^2 (1-\beta)^2} + \frac{2(1-\delta_2)HL^2}{n} \sum_{i\in[n]} \sum_{\tau=t_0+1}^{t-1} \mathbb{E} \|p_{i,\tau}\|^2$$

$$\leq \frac{2(1-\delta_2)(1-\delta_1)\eta^2 H^2 L^2 V_2}{\left(1-\sqrt{1-\delta_1}\right)^2 (1-\beta)^2} + \frac{2(1-\delta_2)\eta^2 H^2 L^2 V_2}{(1-\beta)^2} \qquad \triangleright \text{Lemma 4}$$

$$\leq \left( \frac{1-\delta_1}{\left(1-\sqrt{1-\delta_1}\right)^2} + 1 \right) \frac{2(1-\delta_2)\eta^2 H^2 L^2 V_2}{(1-\beta)^2}.$$

Finally, combining all the ingredients above, we have

$$\mathbb{E}\left[ F(\bar{z}_t) - F(\bar{z}_{t-1}) \right]$$

$$\leq \text{①} + \frac{L}{2}\text{②}$$

$$= \text{③} + \text{④} + \frac{L}{2} \left[ \frac{\eta^2}{n(1-\beta)^2} V_1 + \frac{\eta^2}{(1-\beta)^2} \mathbb{E} \left\| \frac{1}{n} \sum_{i\in[n]} \nabla F_i(x_{i,t-1}) \right\|^2 \right]$$

$$\leq \frac{1}{2(1-\beta)}\text{⑤} + \frac{\eta^2}{2(1-\beta)} \mathbb{E} \left\| \frac{1}{n} \sum_{i\in[n]} \nabla F_i(x_{i,t-1}) \right\|^2$$

$$- \frac{\eta}{2(1-\beta)} \mathbb{E} \|\nabla F(\bar{x}_{t-1})\|^2 - \frac{\eta}{2(1-\beta)} \mathbb{E} \left\| \frac{1}{n} \sum_{i\in[n]} \nabla F_i(x_{i,t-1}) \right\|^2 + \frac{\eta}{2(1-\beta)}\text{⑥}$$

$$+ \frac{L}{2} \left[ \frac{\eta^2}{n(1-\beta)^2} V_1 + \frac{\eta^2}{(1-\beta)^2} \mathbb{E} \left\| \frac{1}{n} \sum_{i\in[n]} \nabla F_i(x_{i,t-1}) \right\|^2 \right]$$

$$\leq \frac{1}{2(1-\beta)} \frac{L^2 \eta^2 \beta^4}{(1-\beta)^4} V_2 + \frac{\eta^2}{2(1-\beta)} \mathbb{E} \left\| \frac{1}{n} \sum_{i\in[n]} \nabla F_i(x_{i,t-1}) \right\|^2$$

$$- \frac{\eta}{2(1-\beta)} \|\nabla F(\bar{x}_{t-1})\|^2 - \frac{\eta}{2(1-\beta)} \mathbb{E} \left\| \frac{1}{n} \sum_{i\in[n]} \nabla F_i(x_{i,t-1}) \right\|^2$$

$$+ \frac{\eta}{2(1-\beta)} \left( \frac{1-\delta_1}{\left(1-\sqrt{1-\delta_1}\right)^2} + 1 \right) \frac{2(1-\delta_2)\eta^2 H^2 L^2 V_2}{(1-\beta)^2}$$

$$+ \frac{L}{2} \left[ \frac{\eta^2}{n(1-\beta)^2} V_1 + \frac{\eta^2}{(1-\beta)^2} \mathbb{E} \left\| \frac{1}{n} \sum_{i\in[n]} \nabla F_i(x_{i,t-1}) \right\|^2 \right]$$

$$\leq \frac{1}{2(1-\beta)} \frac{L^2 \eta^2 \beta^4}{(1-\beta)^4} V_2 + \frac{L}{2} \frac{\eta^2}{n(1-\beta)^2} V_1 - \frac{\eta}{2(1-\beta)} \|\nabla F(\bar{x}_{t-1})\|^2$$

$$\triangleright \text{using } \eta \leq \min\{\tfrac{1}{2}, \tfrac{1-\beta}{2L}\}$$

$$+ \frac{\eta}{2(1-\beta)} \left( \frac{1-\delta_1}{\left(1-\sqrt{1-\delta_1}\right)^2} + 1 \right) \frac{2(1-\delta_2)\eta^2 H^2 L^2 V_2}{(1-\beta)^2}$$

$$\leq \frac{\eta^2 \beta^4 L^2 V_2}{2(1-\beta)^5} + \frac{\eta^2 L V_1}{2n(1-\beta)^2} - \frac{\eta}{2(1-\beta)} \|\nabla F(\bar{x}_{t-1})\|^2$$

$$+ \left( \frac{1-\delta_1}{\left(1-\sqrt{1-\delta_1}\right)^2} + 1 \right) \frac{(1-\delta_2)\eta^3 H^2 L^2 V_2}{(1-\beta)^3}.$$

By re-arranging the terms, we have

$$\|\nabla F(\bar{x}_{t-1})\|^2$$

$$\leq \frac{2(1-\beta)\mathbb{E}\left[F(\bar{z}_{t-1}) - F(\bar{z}_t)\right]}{\eta} + \frac{\eta^2 \beta^4 L^2 V_2}{(1-\beta)^4} + \frac{\eta L V_1}{n(1-\beta)}$$

$$+ \left( \frac{1-\delta_1}{\left(1-\sqrt{1-\delta_1}\right)^2} + 1 \right) \frac{2(1-\delta_2)\eta^2 H^2 L^2 V_2}{(1-\beta)^2}.$$

By telescoping and taking total expectation, after $T$ iterations, we have

$$\frac{1}{T} \sum_{t=1}^{T} \mathbb{E}\left[\|\nabla F(\bar{x}_{t-1})\|^2\right]$$

$$\leq \frac{2(1-\beta)\left[F(\bar{x}_0) - F(x_*)\right]}{\eta T} + \frac{\eta^2 \beta^4 L^2 V_2}{(1-\beta)^4} + \frac{\eta L V_1}{n(1-\beta)}$$

$$+ \left( \frac{1-\delta_1}{\left(1-\sqrt{1-\delta_1}\right)^2} + 1 \right) \frac{2(1-\delta_2)\eta^2 H^2 L^2 V_2}{(1-\beta)^2}$$

□

**Corollary 2.** *Taking* $\eta = \min\left\{ \frac{\gamma}{\sqrt{T/n} + \left[\left(4(1-\delta_1)/\delta_1^2+1\right)2(1-\delta_2)H^2+1\right]^{1/3}T^{1/3}}, \frac{1}{2} \right\}$ *for some* $\gamma > 0$*, after* $T \geq \frac{4\gamma^2 L^2 n}{(1-\beta)^2}$ *iterations, Algorithm 4 (CSERM) converges to a critical point:* $\frac{1}{T}\sum_{t=1}^{T} \mathbb{E}\left[\|\nabla F(\bar{x}_{t-1})\|^2\right] \leq \frac{1}{\sqrt{nT}}\left[\frac{2(1-\beta)[F(\bar{x}_0)-F(x_*)]}{\gamma} + \frac{\gamma L V_1}{(1-\beta)}\right] + \frac{\left[\left(4(1-\delta_1)/\delta_1^2+1\right)2(1-\delta_2)H^2+1\right]^{1/3}}{T^{2/3}}\left[\frac{2(1-\beta)[F(\bar{x}_0)-F(x_*)]}{\gamma} + \frac{\gamma^2 L^2 V_2}{(1-\beta)^4}\right].$

*Proof.* From Theorem 2, taking $\eta \leq \min\{\frac{1}{2}, \frac{1-\beta}{2L}\}$, we have

$$\frac{1}{T} \sum_{t=1}^{T} \mathbb{E}\left[\|\nabla F(\bar{x}_{t-1})\|^2\right]$$

$$\leq \frac{2(1-\beta)\left[F(\bar{x}_0) - F(x_*)\right]}{\eta T} + \frac{\eta^2 \beta^4 L^2 V_2}{(1-\beta)^4} + \frac{\eta L V_1}{n(1-\beta)}$$

$$+ \left( \frac{4(1-\delta_1)}{\delta_1^2} + 1 \right) \frac{2(1-\delta_2)\eta^2 H^2 L^2 V_2}{(1-\beta)^2}.$$

Denote $F_o = 2\left[F(\bar{x}_0) - F(x_*)\right]$, and $C = \left[\frac{4(1-\delta_1)}{\delta_1^2} + 1\right]2(1-\delta_2)H^2 + 1$. Taking $\eta = \min\left\{ \frac{\gamma}{\sqrt{T/n} + C^{1/3}T^{1/3}}, \frac{1}{2} \right\}$, and $T \geq \frac{4\gamma^2 L^2 n}{(1-\beta)^2}$, we have $\eta \leq \frac{1-\beta}{2L}$. Thus, we have

$$\frac{1}{T} \sum_{t=1}^{T} \mathbb{E}\left[\|\nabla F(\bar{x}_{t-1})\|^2\right]$$

$$\leq \frac{(1-\beta)F_o}{\eta T} + \frac{L\eta V_1}{n(1-\beta)} + \frac{C\eta^2 L^2 V_2}{(1-\beta)^4}$$

$$\leq \frac{(1-\beta)F_o}{T} \frac{\sqrt{T/n} + C^{1/3}T^{1/3}}{\gamma} + \frac{LV_1}{n(1-\beta)} \frac{\gamma}{\sqrt{T/n}} + CL^2 V_2 \frac{\gamma^2}{C^{2/3}T^{2/3}(1-\beta)^4}$$

$$\leq \frac{(1-\beta)F_o}{\gamma\sqrt{nT}} + \frac{(1-\beta)F_o C^{1/3}}{\gamma T^{2/3}} + \frac{C^{1/3}\gamma^2 L^2 V_2}{T^{2/3}(1-\beta)^4} + \frac{\gamma LV_1}{(1-\beta)\sqrt{nT}}$$

$$\leq \frac{1}{\sqrt{nT}} \left[ (1-\beta)\frac{F_o}{\gamma} + \frac{\gamma LV_1}{(1-\beta)} \right] + \frac{C^{1/3}}{T^{2/3}} \left[ \frac{(1-\beta)F_o}{\gamma} + \frac{\gamma^2 L^2 V_2}{(1-\beta)^4} \right].$$

$\square$

## C Compressor configurations

In Table 3, we show the best configurations of the hyperparameters $H$, $R_{\mathcal{C}_1}$, and $R_{\mathcal{C}_2}$ for each optimizer and overall compression ratio $R_{\mathcal{C}}$. When tuning the hyperparameters, given the overall compression ratio $R_{\mathcal{C}}$, we enumerate the hyperparameters that satisfies $R_{\mathcal{C}} = \frac{1}{1/R_{\mathcal{C}_2} + 1/(R_{\mathcal{C}_1} \times H)}$, such that $H \geq 2$, $R_{\mathcal{C}_1} \geq 1$, and $R_{\mathcal{C}_2} \geq 4$ are all varied in $\{2^c : c \in \{0, 1, \ldots, 10\}\}$

Table 3: Compressor configurations

| Optimizer | Overall $R_{\mathcal{C}}$ | $R_{\mathcal{C}_2}$ | $R_{\mathcal{C}_1}$ | H |
|---|---|---|---|---|
| EF-SGD | 2 | | 2 | |
| QSparse-local-SGD | 2 | | 1 | 2 |
| CSEA | 2 | | 2 | |
| CSER | 2 | 4 | 2 | 2 |
| EF-SGD | 4 | | 4 | |
| QSparse-local-SGD | 4 | | 1 | 4 |
| CSEA | 4 | | 4 | |
| CSER | 4 | 8 | 2 | 4 |
| CSER-PL | 4 | | 2 | 2 |
| EF-SGD | 8 | | 8 | |
| QSparse-local-SGD | 8 | | 1 | 8 |
| CSEA | 8 | | 8 | |
| CSER | 8 | 16 | 2 | 8 |
| CSER-PL | 8 | | 2 | 4 |
| EF-SGD | 16 | | 16 | |
| QSparse-local-SGD | 16 | | 4 | 4 |
| CSEA | 16 | | 16 | |
| CSER | 16 | 32 | 8 | 4 |
| CSER-PL | 16 | | 4 | 4 |
| EF-SGD | 32 | | 32 | |
| QSparse-local-SGD | 32 | | 4 | 8 |
| CSEA | 32 | | 32 | |
| CSER | 32 | 64 | 8 | 8 |
| CSER-PL | 32 | | 8 | 4 |
| EF-SGD | 64 | | 64 | |
| QSparse-local-SGD | 64 | | 16 | 4 |
| CSEA | 64 | | 64 | |
| CSER | 64 | 128 | 8 | 16 |
| CSER-PL | 64 | | 8 | 8 |
| EF-SGD | 128 | | 128 | |
| QSparse-local-SGD | 128 | | 16 | 8 |
| CSEA | 128 | | 128 | |
| CSER | 128 | 256 | 4 | 64 |
| CSER-PL | 128 | | 8 | 16 |
| EF-SGD | 256 | | 256 | |
| QSparse-local-SGD | 256 | | 128 | 2 |
| CSEA | 256 | | 256 | |
| CSER | 256 | 512 | 16 | 32 |
| CSER-PL | 256 | | 16 | 16 |
| EF-SGD | 512 | | 512 | |
| QSparse-local-SGD | 512 | | 128 | 4 |
| CSEA | 512 | | 512 | |
| CSER | 512 | 1024 | 8 | 128 |
| CSER-PL | 512 | | 16 | 32 |
| EF-SGD | 1024 | | 1024 | |
| QSparse-local-SGD | 1024 | | 128 | 8 |
| CSEA | 1024 | | 1024 | |
| CSER | 1024 | 2048 | 32 | 64 |
| CSER-PL | 1024 | | 32 | 32 |

# D  Additional experiments

In this section, we present additional experiments, including the results on the special cases CSEA and CSER-PL. For CIFAR-100, we also report the results on relatively small overall compression ratios ($R_{\mathcal{C}} \in \{2, 4, 8\}$). The training loss vs. the number of epochs is also reported for both CIFAR-100 and ImageNet. Furthermore, we report the testing accuracy vs. the communication overhead (in bits).

(a) $CR = 32$, accuracy vs. # epochs

(b) $CR = 256$, accuracy vs. # epochs

(c) $CR = 1024$, accuracy vs. # epochs

Figure 3: Testing accuracy vs.# of epochs with different algorithms, for WideResNet-40-8 on CIFAR-100.

(a) $CR = 32$, accuracy vs. training time

(b) $CR = 256$, accuracy vs. training time

(c) $CR = 1024$, accuracy vs. training time

Figure 4: Testing accuracy vs. training time with different algorithms, for WideResNet-40-8 on CIFAR-100.

(a) $CR = 32$, accuracy vs. communication

(b) $CR = 256$, accuracy vs. communication

(c) $CR = 1024$, accuracy vs. communication

Figure 5: Testing accuracy vs. communication with different algorithms, for WideResNet-40-8 on CIFAR-100.

(a) $CR = 32$, loss vs. # epochs

(b) $CR = 256$, loss vs. # epochs

(c) $CR = 1024$, loss vs. # epochs

Figure 6: Training loss vs.# of epochs with different algorithms, for WideResNet-40-8 on CIFAR-100.

(a) $CR = 32$, accuracy vs. # epochs

(b) $CR = 256$, accuracy vs. # epochs

(c) $CR = 1024$, accuracy vs. # epochs

Figure 7: Testing accuracy vs.# of epochs with different algorithms, for ResNet-50 on Imagenet.

(a) $CR = 32$, accuracy vs. training time

(b) $CR = 256$, accuracy vs. training time

(c) $CR = 1024$, accuracy vs. training time

Figure 8: Testing accuracy vs. training time with different algorithms, for ResNet-50 on Imagenet.

(a) $CR = 32$, accuracy vs. communication

(b) $CR = 256$, accuracy vs. communication

(c) $CR = 1024$, accuracy vs. communication

Figure 9: Testing accuracy vs. communication with different algorithms, for ResNet-50 on Imagenet.

(a) $CR = 32$, loss vs. # epochs

(b) $CR = 256$, loss vs. # epochs

(c) $CR = 1024$, loss vs. # epochs

Figure 10: Training loss vs.# of epochs with different algorithms, for ResNet-50 on Imagenet.