[Reviews · NeurIPS 2020]

Review 1

Summary and Contributions: The submission presents CSER, a new take on communication-compressed data-parallel SGD. In brief, the main idea here is a tweak of QSparse-local-SGD--which is one of the top methods for reducing the distribution overheads of SGD, which combined local-SGD (infrequent sync) with communication compression (gradient sparsification)—by allowing the local models at the processors to incorporate the compression errors. This reduces the “gradient staleness” in the system—in some sense—and allows the algorithm to converge more accurately than QSparse. Empirical results on image classification models (WRN and RN50 on CIFAR-100/ILSVRC) are shown to back this up.

Strengths: - The paper introduces a reasonable extension of previous work. - This idea is well investigated via by analysis, and largely through experiments. - Paper is well-written and easy to follow. Thanks for that. - Some aspects of the empirical validation (e.g. number of repetitions) are very well executed.

Weaknesses: - One could say that the main idea is a fairly limited tweak relative to previous work. The results are well-executed but ultimately not very surprising. - I’m having trouble checking the exact accuracy results for the RN50 experiments, which don’t seem to appear in any table in the text or additional material. The paper claims “no accuracy loss,” but I can’t really verify that, and, as far as I can tell visually, in some of the RN50 experiments, there is some accuracy loss.

Correctness: The claims are, to my reading, correct. The empirical methodology is good, but some of the result reporting can be improved.

Clarity: Yes.

Relation to Prior Work: The previous work is covered well, although the authors might want to position themselves also relative to the work on decentralized SGD (https://arxiv.org/abs/1902.00340), which is using similar ideas.

Reproducibility: Yes

Additional Feedback: Comments: - There are a bunch of minor typos. - How is the experimental setup chosen? The 8 nodes with a V100 each is somewhat unusual—in a cloud setup, one would rather instantiate 8 V100s in the same chassis, reducing communication and latency cost. One could suspect the authors of artificially trying to increase communication latency and bandwidth cost this way, so maybe it would be a good idea to motivate this better, or show results for a single-node multi-GPU case. - What is your baseline exactly? is it using NCCL, or just some version of MPI? - You really need to provide exact accuracy results for the Imagenet experiments. The WRN model you are using is clearly overparametrized, so it is not surprising that it can withstand a ton of compression tricks without losing accuracy. Ultimately your claimed advantage over QSparse is in terms of accuracy per bit, so you need to argue that in a more complex setting. - It would have been nice to have another task as well, e.g. WMT/Transformer. - Your speedup of e.g. 4.5x is reported relative to a baseline that’s not well described, on a setup that’s also a bit hazy. The speedups relative to e.g. QSparse are much lower. === Post-rebuttal comments: In my view, the main shortcoming of the paper is the setting chosen for the experiments, which seems to be chosen precisely so that it provides speedups, as well as the speedup values quoted which are somewhat deceiving. The rebuttal did little to address this issue, since it basically stated that it is similar to other settings considered in the literature. To be precise, in any cloud setting (e.g. AWS) the price per GPU node (e.g. V100 in their case) is the same whether there's one GPU per node (p3.2xlarge), or 8 (p3.16xlarge). Thus, any reasonable person would spin up a machine with 8 GPUs inside a single node, rather than 8 nodes with one GPU each, especially since the 8-node variant would cost more because of storage and network costs, for much poorer bandwidth. I fully understand that intermediate settings exist (2 nodes with 4GPUs each), but it seems to me like the authors chose specifically the extreme setting (8x 1 node per GPU) which would help their results. I frankly am not aware of any realistic setting which would match this, except maybe for supercomputing, where there's indeed one GPU per node (but then the nodes' bandwidth would be much higher). OK, so why does this matter? It matters because their speedups are valid only in the setting where the bandwidth is extremely constrained, and communication costs are high. Otherwise, existing techniques would do just as well. I also share the other reviewers' concern about hyper-parametrization, which is an additional hidden cost. As such, I will maintain my score, but I won't strongly fight against acceptance.


Review 2

Summary and Contributions: This paper proposes a communication-efficient SGD algorithm with error reset, which uses a partial synchronization for both the gradients and the models. Moreover, it prove that the proposed algorithm reaches an improved convergence rate of O(1/\sqrt(nT)), leading a linear speedup using more workers. Extensive experimental results demonstrate the efficiency of the proposed algorithm, such as accelerate the training by nearly 10× for CIFAR-100, and by 4.5× for ImageNet.

Strengths: This paper proposes a novel communication-efficient SGD algorithm by using error reset, which reaches an improved convergence rate of O(1/\sqrt(nT)), leading a linear speedup using more workers. At the same time, extensive experimental results demonstrate the efficiency of the proposed algorithm, such as accelerate the training by nearly 10× for CIFAR-100, and by 4.5× for ImageNet.

Weaknesses: The paper should detail advantage of the error reset, which is a key technique proposed and used in the paper.

Correctness: All methods in the paper are correct.

Clarity: This paper is well written.

Relation to Prior Work: The related work part of the paper at detail discuss the differences between this work and the previous work.

Reproducibility: Yes

Additional Feedback: Some comments are given as follows: Q1: For CSER algorithm, under what circumstances do we need such high compression ratios as 256-1024? Q2: For CSER algorithm, can we apply other compressors except for sparsifier? Q3: Can this CSER algorithm extend non-i.i.d. setting ? Q4: In the CSER and M-CSER algorithms, how to choose two efficient compressors to reach good theoretical results and empirical results ? Q5: In the M-CSER algorithm, I think if you choose a good parameter $\beta$, it will obtain a better convergence rate than the CSER algorithm ? Q6: From the theoretical analysis, the given learning rate $\eta$ is very small. But how to choose the learning rate in the experiments. %%%%%%%%%%%%%%%%%%%%%% Thanks for your responses. I still maintain earlier positive review and recommend acceptance.


Review 3

Summary and Contributions: 1. This paper proposes a novel SGD variant: Communication-efficient SGD with Error Reset, or CSER. 2. The key idea in CSER is a new technique called error reset that corrects for the local model using the compression errors. 3. Partial synchronization leverages advantages from both gradient and model synchronizations.

Strengths: Theoretical grounding: This paper proposes Communication-efficient SGD with Error Reset, and provides convergence analysis. Theoretically, they show that the proposed algorithm enjoys a linear speedup using more workers. Empirical evaluation: Empirically show that with appropriate compression ratios, CSER: i) causes no loss of accuracy, and ii) accelerates training by nearly 10x for CIFAR-100, and by 4.5x for ImageNet. by utilizing both gradient and model synchronization, and balancing the communication budget between them, CSER achieves a better trade-off between the accuracy and the reduction of bidirectional communication.

Weaknesses: 1. Experimental results are only reported on CIFAR-100 (Wide-ResNet-40-8) and ImageNet (ResNet-50). What are the impact of different network structures, such as DenseNet, ShuffleNet[1]? 2. In Section 5.2, the author says "Note that for CSER and QSparse-local-SGD with the same overall RC, the configuration of H, RC1, and RC2 is not unique. We try multiple configurations and report the ones perform best on the training loss". The configuration of H, RC1, and RC2 is reported in Appendix C when training loss is best. Is there any relationship between training loss and the configuration of H, RC1, and RC2? 3. Is there any influence from the number of machines in a cluster on the testing accuracy with different compression ratios? [1] ShuffleNet: An Extremely Efficient Convolutional Neural Network for Mobile Devices. Xiangyu Zhang, Xinyu Zhou, Mengxiao Lin, Jian Sun

Correctness: Yes

Clarity: The paper is generally well-written and structured clearly.

Relation to Prior Work: The Related Work section is well-written. Good job!

Reproducibility: Yes

Additional Feedback: UPDATE: After reading the other reviews and the author feedback, I still maintain my score, because of the new take on communication-compressed data-parallel SGD.


Review 4

Summary and Contributions: This paper introduces an algorithm to compress the gradients in distributed SGD to mitigate the communication-bottleneck. Building on error feedback and local SGD, the algorithm splits every gradient into two parts: one that can be readily compressed and averaged using a compressor A, and a remainder. Both parts are directly used to update the worker's local models, but the 'remainders' are saved up to be averaged with another compressor B at more infrequent intervals. The paper includes convergence proofs for the proposed method and experiments that show convergence on Cifar-100 and ImageNet with extreme communication compression (1000x) within a standard epoch budget.

Strengths: The method is well motivated and shows promising results. The combination of existing components for this application is novel.

Weaknesses: I interpret the method as introducing two changes compared to EF-SGD and Qsparse-SGD: (1) models are not forced to remain synchronized and can move faster, and (2) two types of averaging at different frequencies. It is intuitive to me why (1) helps, but I am less convinced about (2). To assert that this seemingly complicated two-level scheme is useful, I would like to see H=1 as a baseline in the main paper. It could be that these results are already available in the Appendix, but I couldn't quickly tell where to look. The results in the paper focus on comparing different algorithms at their best configuration (allocation of communication budget between two compressors and H). Because the methods are strongly related, I would find it more interesting to see how the communication allocationtrade-off between the two compressors and the averaging interval H affect performance of the proposed method. I also see the introduction of two compressors vs one as a yet another hyperparameter to tune and a minor weakness. I can accept this if motivation for this two-level scheme is made stronger.

Correctness: I see no problems in the correctness of this work.

Clarity: The paper was pleasant to read and has few errors. Some comments: - The speedup numbers in the abstract do not mean much without context about the setup. - I find the term 'error reset' slightly misleading. In my understanding, the errors are also slowly averaged, and not reset to 0. And very minor: - line 33: fail -> fails

Relation to Prior Work: The authors position their work clearly with respect to Q-sparse SGD, EF-SGD and local SGD. A few notes: - The statement that "The state-of-the-art work in communication-efficient SGD is called QSParse-local-SGD" (line 20) strikes me as too bold. Recent reviews like (https://repository.kaust.edu.sa/bitstream/handle/10754/662495/gradient-compression-survey.pdf?sequence=1) do not draw such clear-cut conclusions. - The aspect of this algorithm that local models are not kept synchronized appears frequently in decentralized learning. Except from the two-level synchronization scheme, I believe the algorithm has strong parallels with ChocoSGD [Koloskova et al. 2019] and DeepSqueeze [Tang et al. 2019] - The block-wise random-k compression with synchronized randomness that the authors propose also appears in PowerSGD [Vogels et al. 2019].

Reproducibility: Yes

Additional Feedback: My main concern with this submission is the lack of motivation for the two-level averaging with different compressors and different intervals. Experiments are needed to strongly motivate the need for this complication. If the authors can point me to those experiments in the Appendix, I may increase my score. -- Edit: thank you for pointing me to Table 4. There indeed seems to be a benefit of the two-level scheme. Whether the benefits warrant the additional complexity of the two-level scheme, probably depends on the use case. It is still not very intuitive to me why the two compressors help. I would recommend to put more emphasis on this question in the paper.

[Author Response · NeurIPS 2020]

**Reviewer #1:**

**"limited tweak relative to previous work"** The contributions of CSER put it beyond merely a "tweak". Our two key
contributions are: (1) A novel mechanism "error reset" that uses arbitrary compressors in a way different from error
feedback. Our theoretical analysis (Theorem 1, Lemma 2, Remark 1,2) shows that error reset achieves better error
bounds than error feedback. Empirical results show better convergence of error reset, especially for high compression
ratios. (2) New combination of partial gradient and model synchronization–in particular, by carefully distributing the
communication budgets between two synchronizations, we can further improve the convergence. As shown in Table 4
in Appendix P30, CSER combining 2 synchronizations works better than only using one of them (CSEA) in most cases.
**"results not very surprising"** Our work is the first to push the compression ratio to 1024 for both worker-to-server and
server-to-worker communication, where CSER shows much better convergence than previous work. Particularly, at
points when EF-SGD and QSparse diverge, CSER converges well. So far, the experiments with largest compression ratio
were proposed by Deep Gradient Compression [Lin et al., 2018], which only considers worker-to-server communication
with the compression ratio $\approx 600$, and it had no theoretical analysis. Our GRBS compressor (Section 3.3) reduces
bidirectional communication with the desired high compression ratio ($\geq 256$), which is not explored in previous work.
**"exact accuracy results for the Imagenet"** See the table below.

**"No accuracy loss"** is conditional. For both CIFAR
and ImageNet, the accuracy is slightly better than full-
precision SGD when $R_C \leq 16$. With larger compression
ratios, CSER has some accuracy loss compared to full-
precision SGD, but performs much better than EF-SGD
and QSparse. We will revise the claim by adding context.

| $R_C$/Optimizer | 1 | 16 | 32 | 256 | 1024 |
|---|---|---|---|---|---|
| SGD | $76.41_{\pm 0.03}$ | | | | |
| EF-SGD | | $76.34_{\pm 0.06}$ | $76.19_{\pm 0.07}$ | $69.73_{\pm 0.66}$ | diverge |
| QSparse | | $76.40_{\pm 0.05}$ | $73.89_{\pm 0.09}$ | diverge | diverge |
| CSER | | $\mathbf{76.53}_{\pm 0.05}$ | $\mathbf{76.33}_{\pm 0.06}$ | $\mathbf{75.94}_{\pm 0.09}$ | $\mathbf{74.93}_{\pm 0.11}$ |

**"is it using NCCL"** All the algorithms use the same communication library: Horovod with NCCL.
**"How is the experimental setup chosen"** Multiple nodes connected by 10Gb/s Ethernet is a typical setup used in
previous works such as [Lin et al., 2018], signSGD [Bernstein et al., 2019] and EF-SGD [32]. When using single node
with multiple GPUs connected by NVLink, the communication will be extremely fast and compression is less necessary.
In this work, we aim to show that we can significantly reduce the heavy inter-node communication. Indeed, one may
increase number of GPUs per node to do large batch training, but this becomes prohibitively expensive due to GPU cost.
**"WMT/Transformer"** typically uses SGD variants with adaptive learning rates such as ADAM. In this paper we focus
on SGD with momentum without adaptive learning rates. Applying error reset to ADAM is future work.
**"speedups relative to QSparse"** CSER, EF-SGD and QSparse use exactly the same amount of communication, thus
theoretically having the same training time with the same overall $R_C$. The advantage over QSparse is that CSER
converges much better with the same low amount of communication. Figure 1(e), 2(d) show slightly shorter training
time of CSER because of less memory copy in computation, which is irrelevant to communication overhead.

**Reviewer #2:**

**"High compression ratios"** are useful when network bandwidth is very low and model sizes are very big.
**"other compressors"** Yes. Our theoretical analysis works for arbitrary compressors.
**"non-i.i.d."** Yes. Our theoretical analysis already applies to non-iid case. In Assumption 2, we do not assume identical
workers. In our proof (line 404 in Appendix), we only need independence to obtain $V_1/n$ variance.
**how to choose two efficient compressors** Theorem 1 shows how the configurations of compressors affect convergence.
With fixed overall $R_C$, we can enumerate possible configurations (as shown in Table 3 in Appendix P29) to get relatively
smaller error bounds. To find the best configuration in practice, we do grid search.
**"choose a good $\beta$"** is possible, but irrelevant to communication compression. So we just use the common value 0.9.
**"how to choose the learning rate"** We use grid search to tune the learning rate. Details can be found in Section 5.1.
**"detail advantage of the error reset"** We will highlight the advantages of error reset in the revision.

**Reviewer #3:**

**only reported on ResNet** Prior work such as Deep Gradient Compression [Lin et al., 2018], EF-SGD [32] and QSparse
[3], all used CIFAR-10/100 and ImageNet + ResNet in the experiments. We choose the same to directly contrast results.
**"relationship between training loss and the configuration of H, RC1, and RC2"** Theorem 1 shows the relationship
between squared gradient norm and configurations ($R_{C_1} = 1/\delta_1, R_{C_2} = 1/\delta_2$). Though we cannot translate the
convergence rate of gradient norm into the one of training loss due to non-convexity, in practice better convergence on
the gradient norm implies faster convergence on the training loss. With fixed overall $R_C = 1/[1/(R_{C_1} \times H) + 1/R_{C_2}]$,
we can enumerate possible configurations to get relatively smaller error bounds, but the optimal configuration is
unknown. To find the best configuration in practice, we do grid search.
**"influence from the number of machines"** This is identical behavior as full-precision SGD, in that changing the
number of machines affects the global batch sizes, thus affects the testing accuracy and requires different learning rates.

**Reviewer #4:**

**"H=1 as a baseline"** CSER with $H = 1$ is a special case called "CSEA", which is also novel. The results are in
Appendix P30, Table 4 and subsequent figures. CSEA uses only **one compressor**. In Table 4, we can see that CSER
with 2 compressors outperforms CSEA in most cases.
By **"state-of-the-art"** we mean the best latest work combining both local SGD and compression. We will clarify it,
and add decentralized SGD and PowerSGD to the related work.

[Meta-Review · NeurIPS 2020]

Three reviewers agree that this submission represents an important contribution to the field. However, a fourth expressed significant concerns about the empirical evaluation. Please be sure to carefully review and address the concerns of all reviewers in the revision.